# Tailoring the superradiant and subradiant nature of two coherently coupled quantum emitters

J.-B. Trebbia[1,2], Q. Deplano[1,2], P. Tamarat[1,2] & B. Lounis [1,2✉]

The control and manipulation of quantum-entangled states is crucial for the development of quantum technologies. A promising route is to couple solid-state quantum emitters through their optical dipole-dipole interactions. Entanglement in itself is challenging, as it requires both nanometric distances between emitters and nearly degenerate electronic transitions. Here we implement hyperspectral imaging to identify pairs of coupled dibenzanthanthrene molecules, and find distinctive spectral signatures of maximally entangled superradiant and subradiant electronic states by tuning the molecular optical resonances with Stark effect. We demonstrate far-field selective excitation of the long-lived subradiant delocalized state with a laser field tailored in amplitude and phase. Optical nanoscopy of the coupled molecules unveils spatial signatures that result from quantum interferences in their excitation pathways and reveal the location of each emitter. Controlled electronic-states superposition will help deciphering more complex physical or biological mechanisms governed by the coherent coupling and developing quantum information schemes.

[1] Univ Bordeaux, LP2N, F-33405 Talence, France. [2] Institut d'Optique & CNRS, LP2N, F-33405 Talence, France. ✉email: brahim.lounis@u-bordeaux.fr

The manipulation and coherent control of arrays of inter-acting quantum systems are at the heart of the second quantum revolution. Solid-state quantum emitters such as single molecules[1], quantum dots[2,3] and color centers in diamond[4] are promising candidates for the realization of such arrays in quantum networks[5,6] as they can easily be manipulated with light and integrated in quantum photonics devices[7,8]. Entangling the electronic states of coherently interacting solid-state emitters is challenging since it requires both a coupling strength larger than the coherence decay rate, implying nanometric distances between the emitters, and quasi-degenerate optical transitions, detuned by less than their coupling strength. Among the candidates to achieve this goal, quantum dot molecules[9] are formed by coherent tunneling between individual quantum dots separated by a controlled nanometric distance. Yet, in addition to their complexity and cost of fabrication, these materials suffer from a severe drawback inherent to their self-assembling growth procedure: The poor control over size, strain and composition of the constituent quantum dots results in large detuning of their emission energies[9].

Polycyclic aromatic hydrocarbon molecules embedded in well-chosen solid matrices at liquid helium temperatures have proven to be test-bench systems for quantum optics[10–14] and are intrinsically identical. Although promising, the pro-duction of entangled molecular electronic states for quantum technologies[15] remains hindered by the difficulty to fulfill the requirements of space and transition-frequency proximities of the molecules, due to their random spatial distribution in the host crystal and the variety of local matrix environments that span their optical resonances within an inhomogeneous band. Seeking two molecules matching these coincidences within in a solid matrix is like looking for a needle in a haystack. Coherent optical dipole coupling of a single pair of molecules has been reported only once over the past two decades, in an experiment combining near-field scanning probe microscopy and spectroscopy[16]. Indeed, such local probe techniques are limited in scalability and heavy to implement in cryogenic environments. Further challenges are to manipulate the degree of entanglement in delocalized states of pairs of molecules having frozen geometries and dipole orientations, and selectively address any quantum-entangled state. Coherent and dissipative dipole–dipole interactions give rise to collective phenomena of super- or subradiance[17], in which a collective excitation of the emitters decays faster or slower than the individual molecular excitations, respectively. While superradiance has been widely studied since the pioneering work of Dicke[18], few experimental studies have been reported on subradiance, using essentially ultracold atoms and molecules[19–22], metamaterial lattices[23] or single structured atomic layers acting as optical mirrors[24]. Engineering subradiant states is particularly promising for quantum information storage with real time processing[25–30]. The associated line narrowing may also improve the sensitivity to external fields for quantum metrology applications[31,32].

Here, using a hyperspectral imaging approach to unveil cou-pled pairs of fluorescent molecules in highly doped molecular crystals, we demonstrate the manipulation of the degree of superposition in their electronic states through Stark shifts of their optical resonances. Direct evidence of subradiance (superradiance) is brought through lengthening (shortening) of the fluorescence lifetime recorded when the laser is tuned to the corresponding state. Nearly pure Bell states[33] are achieved and delocalized molecular electronic states are found to extend over distances as large as 60 nm, which opens up attractive perspec-tives in terms of addressability of the quantum emitters. Inter-estingly, far-field super-resolution optical nanoscopy[34] images of the coupled molecules reveal spatial features that result from quantum interferences in their excitation pathways and reveal the exact locations of each molecule. Furthermore, we develop a method of coherent and selective excitation of the long-lived subradiant state, which paves the way to the realization of quantum information schemes based on subradiance[35–37].

## Results

**Spectral signatures of electronic entangled states.** The host-guest system chosen in the experiment consists of dibenzanthan-threne (DBATT) molecules randomly embedded in a naphthalene film with a thickness less than few micrometers (see Methods). At 2 K, thermal dephasing of their optical coherences vanishes, so that single molecules exhibit sharp zero-phonon lines (ZPL) with a natural linewidth of the order of 20 MHz set by the lifetime of the excited state[38,39]. Such molecules nearly behave like simple two-level systems[40] and present a remarkable photostability, allowing quantum optical measurements at strong excitation intensities[41]. For instance, Rabi oscillations in the fluorescence of an isolated single DBATT molecule submitted to a transient resonant excita-tion demonstrate the possibility to prepare the excited state with a probability 0.9 after a $\pi$-pulse and show that the coherence dephasing time reaches its upper limit given by twice the excited state lifetime (Supplementary Fig. 1). The sample is placed at the top of a gold inter-digitated electrode comb with a 20 μm spacing period in order to apply external electric fields up to the MV m$^{-1}$ range and thus tune the molecular optical resonances by Stark effect[42] (see Fig. 1a and Supplementary Fig. 2). In the search for coupled molecules, hyperspectral fluorescence microscopy is per-formed while the laser frequency is scanned across the ~3-nm-wide inhomogeneous band centered on 618 nm. We then select pairs of spatially unresolved molecules having resonance frequencies within the continuous scan range of our laser (<10 GHz). An optical saturation study is then performed on such pairs to seek the spectral signature of dipole–dipole coupling, which manifests in the excitation spectrum as the onset of a central resonance with increasing excitation intensities, as exemplified in Fig. 1b. This central resonance is located at the exact center of the two ZPLs present in the low intensity spectrum and its peak amplitude has a quadratic dependence on the excitation intensity[16]. Such evolution is the hallmark of two-photon simultaneous excitation of both molecules in the state $|E\rangle = |e_1, e_2\rangle$ from the ground state $|G\rangle = |g_1, g_2\rangle$, where $|g_i\rangle$ and $|e_i\rangle$ ($i = 1, 2$) are the ground and first electronic excited states of the bare molecules, respectively (Fig. 1c). Using this method, we have identified a dozen pairs of coupled molecules with various dipole–dipole configurations and coupling strengths. The characteristic spectral fingerprints of three of them are pre-sented in the fluorescence excitation spectra of Fig. 1d–f, other examples being displayed in Supplementary Fig. 3. Figure 1e recalls the so-called J-aggregate configuration[43], where the orien-tations of the transition dipoles $\hat{\mathbf{d}}_1$, $\hat{\mathbf{d}}_2$ are aligned along the molecular separation direction $\hat{\mathbf{r}}_{12}$, while Fig. 1f evokes the so-called H-aggregate configuration with parallel dipoles nearly orthogonal to $\hat{\mathbf{r}}_{12}$.

The vacuum-induced coherent coupling between two molecules, separated by a distance $r_{12}$ much smaller than their transition wavelength, is given by[44,45] $V = 3\alpha\gamma_0[(\hat{\mathbf{d}}_1 \cdot \hat{\mathbf{d}}_2) - 3(\hat{\mathbf{d}}_1 \cdot \hat{\mathbf{r}}_{12})(\hat{\mathbf{d}}_2 \cdot \hat{\mathbf{r}}_{12})]/4(kr_{12})^3$. In this expression, $\gamma_0$ is the radiative decay rate of the electronic excited states $|e_1\rangle$ and $|e_2\rangle$, $k = n(\omega_1 + \omega_2)/2c = n\omega_0/c$ is the wavenumber associated to the mean value $\omega_0$ of the molecular resonance frequencies $\omega_1$ and $\omega_2$ (detuned by $\Delta = \omega_1 - \omega_2$) in a medium with refraction index n, and $\alpha$ is the combined Debye-Waller/Franck-Condon factor of the molecules representing the fraction of photons emitted on the purely electronic transition. Strong dipole–dipole coupling $|V| > \gamma_0$ can be reached for separation distances as large as tens of nm with

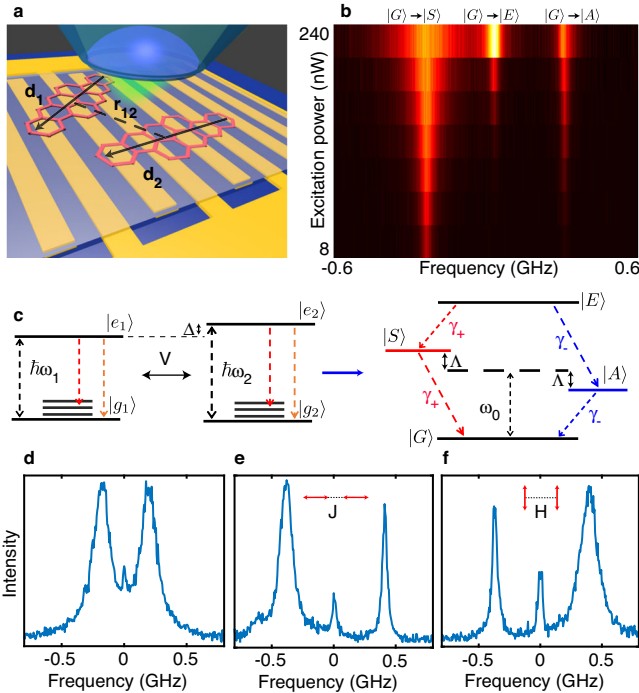

**Fig. 1 Spectral signatures of coherently coupled pairs of molecules.**
**a** Schematic of the sample. A highly doped naphthalene crystal doped with DBATT molecules is placed above a comb of micro-patterned gold electrodes (period of 20 μm) to apply electric fields up to few tens of MV m$^{-1}$. A microscope objective with numerical aperture 0.95 is used to excite the molecules on their ZPLs and collect the red-shifted emitted photons. **b** Series of fluorescence excitation spectra recorded for a pair of coupled molecules at different excitation intensities ranging from 2 W cm$^{-2}$ to 250 W cm$^{-2}$. The onset of a two-photon transition $|G⟩ → |E⟩$ shows direct evidence of coherent dipole–dipole interaction. **c** Left: Energy levels of two bare molecules. Besides the purely electronic transition (orange arrows), the vibrational levels may be involved in the fluorescence process (red arrows) but do not participate to the coherent coupling due to fast relaxation to the ground electronic state (on the picosecond timescale). Right: The coupled molecules are treated as a four-level system with a ground state $|G⟩$, a doubly excited state $|E⟩$ and two split entangled states $|A⟩$ and $|S⟩$. **d–f** Excitation spectra of three different pairs of coupled molecules with various coupling strengths and dipole orientations. The symmetric state $|S⟩$ is associated with in-phase dipoles. In the H-configuration, the electric field created by one dipole is parallel and in-phase opposition with the second dipole, so that the energy shift $-\mathbf{d}_2.\mathbf{E}_1(\mathbf{r}_2)$ of the $|S⟩$ state is positive. In the J-configuration, the electric field is parallel and in phase with the dipole so that the energy shift of the $|S⟩$ state is negative.

nearly parallel transition dipoles and lead to the formation of two entangled delocalized states: A symmetric, superradiant state $|S⟩ = a|g_1, e_2⟩ + b|e_1, g_2⟩$ with enhanced radiative decay rate $\gamma_+ = \gamma_0 + 2ab\gamma_{12}$ and an antisymmetric, subradiant state $|A⟩ = a|e_1, g_2⟩ - b|g_1, e_2⟩$ with reduced radiative decay rate $\gamma_- = \gamma_0 - 2ab\gamma_{12}$ (Fig. 1c). The cross-damping rate $\gamma_{12} = \alpha\gamma_0(\hat{\mathbf{d}}_1.\hat{\mathbf{d}}_2)$ arises from the incoherent coupling of the bare molecules through the vacuum field[44]. The states $|S⟩$ and $|A⟩$ are split by $2\Lambda = \sqrt{\Delta^2 + 4V^2}$ and have projections $a = \sqrt{(\Delta/2 + \Lambda)^2/((\Delta/2 + \Lambda)^2 + V^2)}$ and $b = \sqrt{V^2/((\Delta/2 + \Lambda)^2 + V^2)}$ on the bare states[45], which provide a measure of the degree of electronic state superposition. Maximally entangled states (Bell states $|\Psi_{\text{Bell}}⟩$) corresponding to $a = b = 1/\sqrt{2}$ can be reached when the detuning between the bare molecules is such that $|\Delta| \ll |V|$.

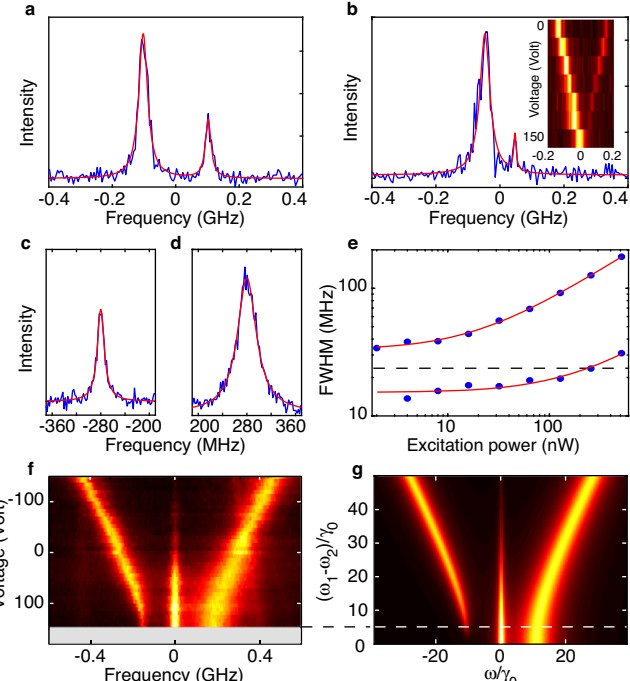

**Fig. 2 Manipulation of the degree of superposition by Stark effect.**
**a**, **b** Excitation spectra of a pair of molecules with nearly parallel dipoles in the J-configuration and $V \sim \gamma_0$, recorded with an excitation intensity of 0.25 W cm$^{-2}$. The applied electric field shifts the molecular detuning from $10\gamma_0$ (**a**) to $4\gamma_0$ (**b**). Lorentzian fits (red curves) of the subradiant (higher-energy) ZPL gives a FWHM linewidth of 22 MHz in (**a**) and 13 MHz in (**b**). The inset in **b** displays the spectral trails of this pair as the voltage is swept from 0 (top) to 150 V (bottom). **c**, **d** Zoomed-in ZPLs of the subradiant (**c**) and the superradiant (**d**) states of another molecular pair with nearly parallel dipoles in the H-dipole configuration and $V \sim 15\gamma_0$. The powers of the Gaussian illumination are respectively 0.1 nW and 0.3 nW. The red curves are Lorentzian fits with FWHM linewidths (13.9 ± 0.6) MHz and (33.0 ± 1.6) MHz, respectively. **e** Saturation plots (blue disks) and fits (red curves) of these ZPL linewidths. Radiative rates $\gamma_-/2\pi = 15.3 ± 0.5$ MHz and $\gamma_+/2\pi = (33.0 ± 0.5)$ MHz are derived from the fitting saturation intensities. The horizontal dashed line at 23 MHz marks the average linewidth of single DBATT molecules at low excitation intensities. **f**, **g** Experimental (**f**) and simulated (**g**) Stark-tuned spectral trails of a coupled pair of molecules under an excitation intensity of 200 W cm$^{-2}$. The experimental spectra have been recentered on the two-photon transition, while the raw spectral trails are presented in Supplementary Fig. 7. The applied voltage is limited to the range −150 to 150 V by electrode breakdown. As the system is tuned towards the anti-crossing point, the two-photon transition gains weight, the superradiant ZPL widens and the subradiant ZPL narrows with vanishing fluorescence amplitude. The simulations are performed for a H-dipole configuration leading to maximal superposition at the anti-crossing point.

**Manipulating the degree of superposition by Stark effect.** Since fluorescent molecules are trapped in the host matrix with fixed positions and orientations that set their coupling strength $V$, the degree of superposition in the states $|S⟩$ and $|A⟩$ can be tuned only by adjusting their resonance frequencies. Manipulation of this degree of superposition by Stark shifting the molecular transitions with a static electric field is demonstrated in Fig. 2 and Supplementary Fig. 4. The spectrum in Fig. 2a presents the two ZPLs of a coupled molecular system (in the J-configuration) consisting of two largely detuned molecules ($|\Delta| \gg |V|$). These lines displaying similar widths and comparable amplitudes arise from the

excitation of nearly localized excited molecular states $|e_1, g_2\rangle$ and $|g_1, e_2\rangle$ ($ab \sim 0$ and $\gamma_+ \sim \gamma_- \sim \gamma_0$). When decreasing $|\Delta|$ (Fig. 2b), the linewidth of the lower-energy, superradiant level broadens while that of the higher-energy, subradiant level sharpens down to 13 MHz. This linewidth is significantly smaller than the natural linewidth $\sim 23$ MHz of uncoupled DBATT molecules embedded in naphthalene[38] (Supplementary Fig. 5) and shows direct evidence for molecular subradiance. Since $\alpha \sim 0.35$ for such molecules (supplementary Fig. 6), this result points to a radiative relaxation rate $\gamma_- \sim \gamma_0(1 - \alpha)$ of a subradiant state formed by two nearly parallel molecular dipoles in the nearly maximal superposition regime. Indeed, the rate $\gamma_-$ reaches its lower bound, as the hallmark of a complete suppression of fluorescence on the purely electronic transition. Interestingly, the reduced energy splitting $2V$ between the two entangled states obtained at the minimal detuning $|\Delta|$ is a signature of a weak dipole–dipole coupling ($V \sim \gamma_0$), which means that the coherent dipole–dipole interaction can create entangled states between solid-state quantum emitters separated by a distance as large as $\sim 60$ nm, using $V = -3\alpha\gamma_0/2(kr_{12})^3$.

Maximal superposition in the states $|S\rangle$ and $|A\rangle$ is also demonstrated for a pair of molecules having nearly parallel transition dipoles in the H-configuration. Figure 2c, d present the fluorescence excitation spectra of their subradiant and superradiant ZPLs recorded in the low saturation regime. The linewidths of these lines $\gamma_-/2\pi \sim (13.9 \pm 0.6)$ MHz and $\gamma_+/2\pi \sim (33.0 \pm 1.6)$ MHz are consistent with their lower and upper bounds $\gamma_\pm \sim \gamma_0(1 \pm \alpha)$. The saturation plots of the ZPL widths presented in Fig. 2e are well reproduced with similar natural linewidths. This confirms the ability to reach the complete delocalization of the excitation over the coupled molecules with maximal superradiance and subradiance effects. Voltage manipulation of the degree of superposition in $|S\rangle$ and $|A\rangle$ is presented in Fig. 2f in the high excitation regime. At large molecular detunings (highest negative voltages), two ZPLs broadened by saturation are observed together with a dim central line assigned to two-photon excitation of $|E\rangle$. As the detuning is reduced, the central line becomes more pronounced and a clear anti-crossing develops between the $|S\rangle$ and $|A\rangle$ levels at the highest positive voltage. The fluorescence intensity of the lowest-energy line vanishes, which reflects the formation of a subradiant antisymmetric state $|A\rangle$ that becomes less efficiently excited with a Gaussian-shaped laser beam centered between the molecules. In contrast, the ZPL associated to the symmetric state $|S\rangle$ broadens due to a more efficient laser excitation.

In order to reproduce these behaviors and characterize the maximal degree of superposition achieved by Stark effect, we compute the fluorescence signal proportional to the sum of the stationary populations of the states $|e_1, g_2\rangle$, $|g_1, e_2\rangle$ and twice that of $|E\rangle$. These populations are derived from the solutions of the Lindblad master equation governed by the Hamiltonian H written in the rotating wave approximation as[45]

$$H = \hbar\omega_0(|E\rangle\langle E| - |G\rangle\langle G|) + \hbar\Lambda(|S\rangle\langle S| - |A\rangle\langle A|)$$
$$- \frac{\hbar}{2}\{[(a\Omega_1 + b\Omega_2)|E\rangle\langle S| + (b\Omega_1 + a\Omega_2)|S\rangle\langle G|]e^{i\omega_L t}$$
$$+ [(a\Omega_2 - b\Omega_1)|E\rangle\langle A| - (b\Omega_2 - a\Omega_1)|A\rangle\langle G|]e^{i\omega_L t} + h.c.\}$$

(1)

where the first two terms describe the coupled molecular system, while the third is its interaction with a driving laser field of frequency $\omega_L$. The Rabi frequencies are defined as $\Omega_i = -\mathbf{d}_i.\mathbf{E}(\mathbf{r}_i - \mathbf{r}_c)/\hbar$, where $\mathbf{d}_i$ ($i = 1,2$) are the molecular transition dipoles and $\mathbf{E}(\mathbf{r}_i - \mathbf{r}_c)$ the complex amplitudes of the laser field at the locations $\mathbf{r}_i$ of the molecules with respect to the laser spot center $\mathbf{r}_c$. The Hamiltonian matrix elements that govern the laser

coupling of the ground state $|G\rangle$ with $|A\rangle$ and $|S\rangle$ are respectively $\langle A|H|G\rangle \propto a\Omega_1 - b\Omega_2$ and $\langle S|H|G\rangle \propto b\Omega_1 + a\Omega_2$. Their dependence on $\Delta$ provide a straightforward understanding of the spectral trails in Fig. 2f, g. For nearly parallel molecular dipoles, a Gaussian-shaped beam centered at mid-distance between the molecules will generate in-phase dipole oscillations with $\Omega_1 = \Omega_2$. When $\Delta$ approaches zero ($a \approx b \approx 1/\sqrt{2}$), the excitation of the symmetric (superradiant) state is therefore enhanced while that of the antisymmetric (subradiant) state becomes forbidden. The simulated spectral trails shown in Fig. 2g quantitatively reproduce the experimental trajectories of Fig. 2f in positions, widths and intensities, considering $V = 10\gamma_0$. At the limit of the voltage breakdown for our electrode array ($\sim 150$ V), the molecular detuning is brought down to $\Delta = 5\gamma_0$ and nearly optimal superposition is achieved with $a = 1.12/\sqrt{2}$ and $b = 0.86/\sqrt{2}$. As defined in Ref. [46] for pure bipartite two-qubit states, the degree of superposition $P_E$ of the two pure states $|S\rangle$ or $|A\rangle$ is obtained from their decomposition $p|\Psi_{Bell}\rangle + \sqrt{1 - p^2}e^{i\phi}|\Psi_{fact}\rangle$ into two orthogonal quantum states, $|\Psi_{Bell}\rangle$ being the corresponding Bell state, $|\Psi_{fact}\rangle$ a factorizable state, $p$ and $\phi$ real numbers. The degree of superposition is defined by $P_E = p^2 = 2|ab|$ and is identical to the concurrence[47]. For the pair of molecules presented in Fig. 2f, $P_E$ reaches ($95 \pm 5$)% at the maximal Stark voltage. Interestingly, this value sets an upper bound to the fidelity achievable when optically preparing the system in the $|S\rangle$ or $|A\rangle$ states from the ground state $|G\rangle$, using a resonant $\pi$-pulse excitation.

The super-and subradiance character of the $|S\rangle$ and $|A\rangle$ states is further evidenced by measurements of their lifetimes for various degrees of superposition. Figure 3a displays the normalized PL decay curves recorded after selective, pulsed excitation of the $|S\rangle$ state (blue curve) and $|A\rangle$ state (red curve) in the situation where the molecular detuning $\Delta$ is minimized by Stark effect. These decays are well reproduced by single exponential curves with lifetimes 6.3 ns and 11.1 ns, respectively, which are markedly different from the average lifetime $7.8 \pm 0.34$ ns of single

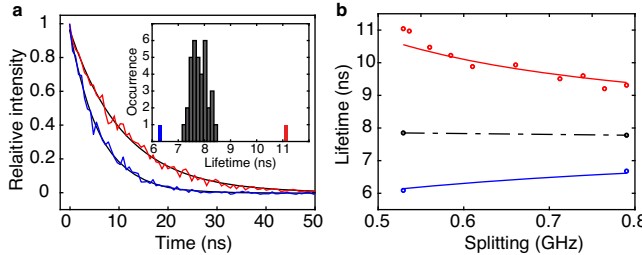

**Fig. 3 Comparison of the subradiant and superradiant lifetimes. a** Decays curves of the subradiant and superradiant states. The laser pulses have a rise-fall time of 1 ns and a repetition rate 100 kHz. The solid curves are exponential fits with lifetimes $\tau_+ = 6.3$ ns and $\tau_- = 11.1$ ns, which are consistent with the computed subradiant lifetime $\tau_- = (\gamma_0 - 2ab\gamma_{12})^{-1} = 1.43\gamma_0^{-1}$ and the superradiant one $\tau_+ = (\gamma_0 + 2ab\gamma_{12})^{-1} = 0.77\gamma_0^{-1}$, using $a = 0.88$, $b = 0.47$, $\gamma_{12} = 0.35\gamma_0$. The coefficients $a$ and $b$ are deduced from the diagonalization of the Hamiltonian $H$ in the absence of laser field. The detuning $\Delta = 15\gamma_0$ is derived from the splitting between $|A\rangle$ and $|S\rangle$ and from the coupling constant $V = 9.5\gamma_0$ that is deduced from the fits of the excitation spectra at various excitation intensities. Inset: Histogram of the lifetime of 35 uncoupled single molecules. The blue and red bars indicate the values of $\tau_+$ and $\tau_-$ for the coupled pair. **b** Evolution of $\tau_+$ (blue circles) and $\tau_-$ (red circles) with the molecular detuning $\Delta$ which is varied by differential Stark shifts of the molecular resonances from $15\gamma_0$ to $32\gamma_0$. The solid curves are the computed values of $\tau_+$ and $\tau_-$. The black circles are the inverse of the average subradiant and superradiant decay rates ($\tau_+^{-1}$ and $\tau_-^{-1}$), and coincide with the average lifetime of the uncoupled single molecules (black dashed lines).

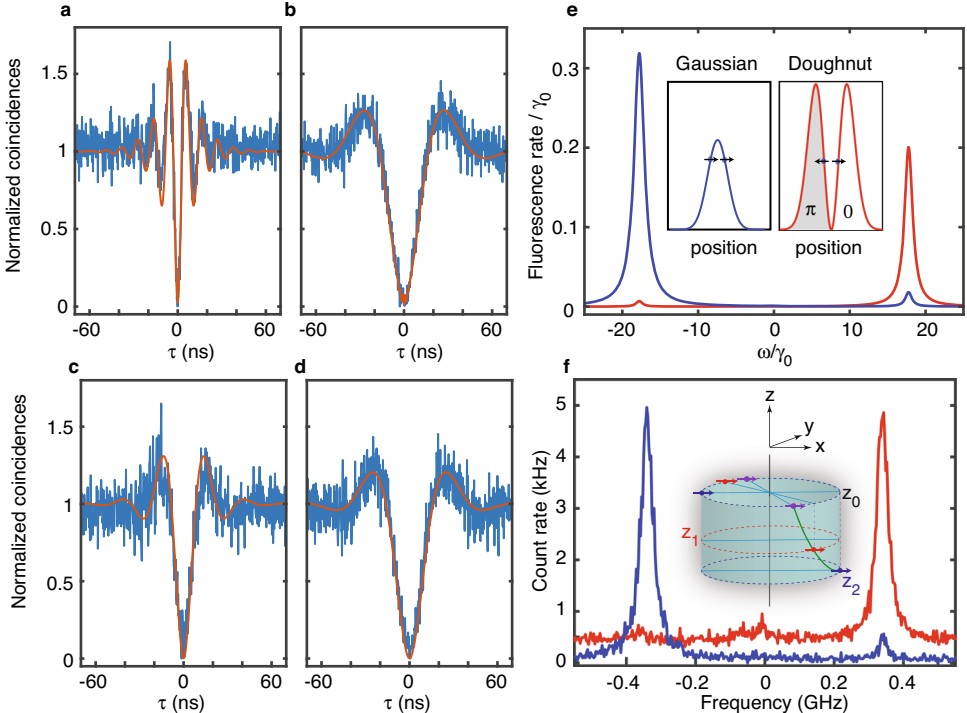

**Fig. 4 Selective excitation of superradiant and subradiant states. a**, **b** Normalized photon coincidence histograms (after background subtraction) measured for a pair of coupled molecules excited with a Gaussian-shaped laser beam of intensity 200 W cm$^{-2}$ at resonance with the $|G\rangle \rightarrow |S\rangle$ transition (**a**) and with the $|G\rangle \rightarrow |A\rangle$ transition (**b**) when the degree of superposition is maximized by Stark effect ($P_E \sim 0.92$, voltage 150 V). They are characterized by an antibunching dip and damped Rabi oscillations, while photon bunching is evidenced when the laser is resonant on the two-photon transition (Supplementary Fig. 10). The second-order correlation functions (red curves) are computed for nearly parallel dipoles in the J-configuration with $V = -17\gamma_0$, $\gamma_{12} = 0.3\gamma_0$, and $\Delta = 10\gamma_0$ (See Methods). **c**, **d** Photon coincidence histograms obtained for the same pair of coupled molecules when the molecular resonances are tuned apart by Stark effect ($-150$ V), leading to a modest degree of superposition $P_E \sim 0.24$. The excitation intensity is 40 W cm$^{-2}$. The simulations are performed with the same parameters except for $\Delta = 60\gamma_0$. **e**, **f** Simulated (**e**) and experimental (**f**) spectra of the same pair excited with a Gaussian-shaped beam of 8 nW (blue curves) and a doughnut-shaped beam of 1 μW (red curves) centered midway between the molecules. The associated profiles of the laser field amplitude and phase are schematically displayed in the inset of (**e**). The simulations are performed with the above values of $V$, $\gamma_{12}$, $\Delta$, and with $a = 0.8$, $b = 0.6$. Depending on whether the emitters are excited in phase (Gaussian beam) or out of phase (doughnut beam), exclusive pumping of the superradiant or the superradiant state, respectively, is performed. The inset of (**f**) exemplifies three parallel-dipole configurations represented by blue, red, and purple couples of arrows, leading to the same coupling strength $V$ and spectra (Supplementary Fig. 11). The positions of the dipoles may differ along the optical (z) axis, while their projections in the focal (x, y) plane are separated by 18 nm.

molecules found in this sample, as shown in the inset of Fig. 3a. One can notice that the average decay rate $(\gamma_+ + \gamma_-)/2$ coincides with the decay rate of the uncoupled single molecules, as a signature of transfer of oscillatory strength between the subradiant and superradiant states. Moreover, we find a good agreement of these lifetimes with the computed values $(\gamma_0 \mp 2ab\gamma_{12})^{-1}$ plotted as red and blue curves in Fig. 3b.

Striking differences in coupling of $|S\rangle$ and $|A\rangle$ to the laser field manifest in the fluorescence intensity autocorrelation function $g^{(2)}(\tau)$ recorded under resonant excitation of $|S\rangle$ and $|A\rangle$ at identical laser intensities. Indeed, besides strong photon antibunching on both transitions, the normalized coincidence histograms show Rabi oscillations that are much faster on the $|G\rangle \rightarrow |S\rangle$ transition (Fig. 4a) than on the $|G\rangle \rightarrow |A\rangle$ one (Fig. 4b) when the degree of superposition in $|S\rangle$ and $|A\rangle$ is maximal. In contrast, the Rabi frequencies become similar when lowering the degree of superposition by Stark shifting these molecular resonances apart (Fig. 4c, d). These correlation functions are well reproduced with numerical simulations, using molecular and laser interaction parameters deduced from prior analysis of the fluorescence excitation spectra, as well as polarization data (See "Methods" and Supplementary Figs. 8 and 9). They are markedly different from the autocorrelation expected for two uncoupled molecules, which would display photon antibunching with a

value $g^{(2)}(0) = 0.5$ at zero-delay time. Incidentally, photon bunching[16] is evidenced when the laser is tuned to the central two-photon resonance (Supplementary Fig. 10), as a signature of simultaneous excitation of coherently coupled molecules to the doubly excited state $|E\rangle$ with the subsequent two-photon emission cascade.

**Selective excitation of super- and subradiant states.** Selective preparation of long-lived entangled states is key for the realization of many quantum information schemes[36,37] and quantum memories[35]. Considering a parallel-dipole configuration, excitation of the subradiant state with a conventional Gaussian-shaped laser field having identical amplitudes and phases at the molecular positions ($\Omega_1 = \Omega_2$) is forbidden according to Eq. (1). However, shaping the field with identical amplitudes and opposite phases at the molecular positions ($\Omega_1 = -\Omega_2$) will enable the excitation of the subradiant state and forbid that of the symmetric superradiant state. We show here that the excitation of the sole antisymmetric subradiant state can be achieved, using a circularly polarized doughnut-shaped (first order Laguerre-Gaussian) beam whose zero-field center is exactly placed midway between both emitters. In the configuration of nearly maximal superposition, selective excitation of either the superradiant or the subradiant

state is theoretically (Fig. 4d) and experimentally (Fig. 4e) demonstrated, using alternatively Gaussian and doughnut types of excitations, respectively. Indeed, the ZPL associated to the $|S\rangle$ state is barely observable in the case of a doughnut beam excitation, while it dominates the $|A\rangle$ line when exciting the system with a Gaussian beam (Fig. 4e). This simple far-field approach is a direct illustration of the ability to tailor selection rules with laser field sculpting. Moreover, it does not suffer from disturbances and quenching effects occurring when the emitters interact with non-uniform near-fields of metallic nanostructures[48].

**Optical nanoscopy of coupled molecules.** While a complete characterization of the molecular and laser interaction parameters is required for a thorough investigation of the superposition signatures, only partial information can be extracted from the above spectroscopic measurements. Indeed, the inset of Fig. 4e illustrates configurations of parallel dipoles with different positions that all lead to reproduce the experimental spectra (Supplementary Fig. 11). Far-field super-resolution imaging techniques allowing nanometric localization of single emitters could be a tool of choice to locate interacting quantum emitters at the nanometer scale. The simplest approach, based on successive localizations of fluorescence spot centers as the laser frequency is tuned[49], will fail at super-localizing the positions of the coupled emitters, since $|S\rangle$ and $|A\rangle$ are spatially delocalized over both molecules. We show in Fig. 5 that the Excited-State Saturation (ESSat) nanoscopy technique[34,50] can reveal the positions of nanometrically spaced coupled emitters through subtle spatial signatures of delocalized quantum states. Using this method, fluorescence images are recorded by scanning over the coupled pair (Fig. 5a) a tightly focused doughnut beam with frequency $\omega_L = (\omega_1 + \omega_2)/2$. At weak saturations, the fluorescence image essentially reflects the intensity distribution of the doughnut-shaped laser beam, and a single fluorescence dip is observed (inset of Fig. 5a). For two *uncoupled* molecules with identical resonance frequencies, the saturation of the molecular transitions would broaden the fluorescence spot with the onset of two sharp dips at the exact positions of the molecules (Supplementary Fig. 12). However, for two coherently coupled molecules subject to different complex Rabi frequencies $\Omega_1 = \Omega(\mathbf{r}_1 - \mathbf{r}_c)$ and $\Omega_2 = \Omega(\mathbf{r}_2 - \mathbf{r}_c)$, non-intuitive features take shape in the ESSat image. Indeed, as the excitation intensity is raised, the central dark spot splits into two fluorescence dips at fixed positions (Fig. 5b). Moreover, two additional dips develop in the direction orthogonal to that of the first ones, with a depth and separation that depend on the intensity (Fig. 5c). Simulations of the fluorescence signals taking into account the vectorial nature of the field at the focal spot[51] reproduce well the experimental ESSat images (Fig. 5b, c) for a pair of highly entangled molecules with parallel transition dipoles. They provide a unique mean to determine the locations of the emitters with a nanometric precision. Indeed, it turns out that none of the fluorescence dips coincides with the actual locations of the molecules, which are located at the extremities of the segment equal and orthogonal to the one connecting the fixed two dips.

In order to interpret these striking image structures on the nanometric scale, we calculate the spatial dependence of eigen-energies of the molecular system coupled to the laser field and the populations of $|G\rangle$, $|S\rangle$, $|A\rangle$ and $|E\rangle$ (Supplementary Note 1). When the doughnut-shaped laser spot is scanned along the inter-molecular x-axis, the fluorescence drops when two dressed levels become degenerate (Fig. 5d). Such coincidences are accompanied by a population increase for $|G\rangle$ and $|A\rangle$ (Fig. 5e). The number of fluorescence dips as well their positions strongly depend on the excitation intensity (Supplementary Note 2 and Supplementary

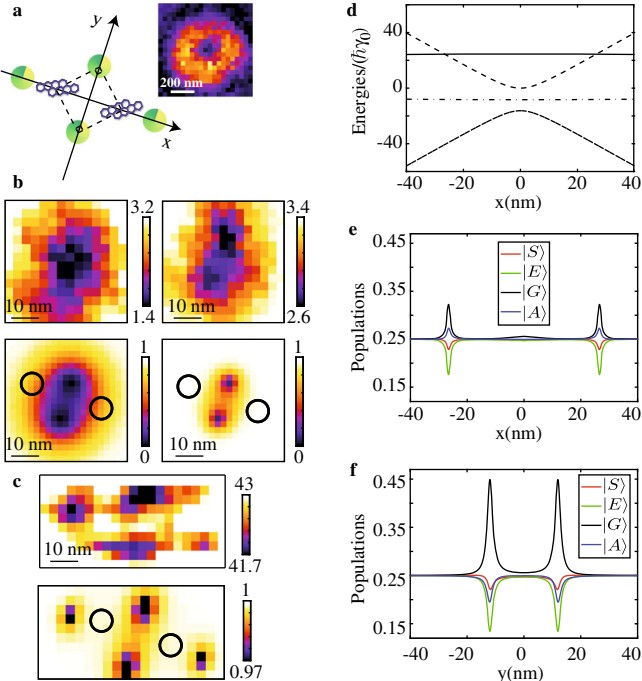

**Fig. 5 Optical nanoscopy of molecules undergoing coherent optical dipole–dipole coupling. a** Configuration of the molecules in the focal plane $(x,y)$. The green disks mark the four positions of the doughnut beam center leading to fluorescence dips. For the two positions marked along the y-axis, the phase difference between the laser fields felt by the two molecules is $\pi/2$. Inset: Fluorescence image recorded at weak saturations. **b, c** Experimental (top frames) and simulated ESSat (bottom frames) images of a pair of coupled molecules excited with a laser tuned on the two-photon transition, with increasing doughnut-peak intensities: 125 W cm$^{-2}$, 500 W cm$^{-2}$ (**b**), $5.4 \times 10^3$ W cm$^{-2}$ (**c**). The simulated ESSat images are computed with $\Delta = 0$, $V = -22\gamma_0$, $\gamma_{12} = 0.5\gamma_0$ and intensities of 175 W cm$^{-2}$, 700 W cm$^{-2}$ (**b**), $7 \times 10^3$ W cm$^{-2}$ (**c**). The in-plane positions of the emitters, separated by $(22 \pm 5)$ nm, are marked with a circle. **d** Evolution of the four eigenenergies of H as the excitation beam is swept along the x-axis. **e** Evolution of the stationary populations of $|G\rangle$,$|A\rangle$, $|S\rangle$ and $|E\rangle$ along the x-axis. Each eigenstate experiences a position-dependent light shift and the fluorescence signal drops at the degeneracy points. Such coincidences are accompanied by a population increase for the $|G\rangle$ and $|A\rangle$. **f** Evolution of the stationary populations of $|G\rangle$, $|A\rangle$, $|S\rangle$ and $|E\rangle$ as the laser spot is swept along the y-axis. The peaks of $|G\rangle$ population occur at the two positions where $\Omega_2 = \Omega_1 e^{\pm \frac{i\pi}{2}}$.

Fig. 13). When the spot is scanned along the perpendicular (y-axis) direction, both emitters undergo the same laser intensity ($|\Omega_1| = |\Omega_2|$) and the fluorescence dips occur at the positions where the phases taken by the laser field at the molecular positions differ by $\pi/2$. Such situations translate into an increase in the population of $|G\rangle$ at the expense of that of the excited states, as shown in Fig. 5f. This can be interpreted as the result of a destructive interference between the two excitation paths which bring the system from $|G\rangle$ to $|E\rangle$: $|G\rangle \rightarrow |A\rangle \rightarrow |E\rangle$ and $|G\rangle \rightarrow |S\rangle \rightarrow |E\rangle$. In the quasi-resonant regime, the two-photon transition can be described by an effective interaction Hamiltonian with a coupling matrix element[52] $\hbar\Omega_{EG}^{\text{eff}} = \frac{\langle E|H|A\rangle\langle A|H|G\rangle}{\hbar(\omega_0 - \omega + \Lambda)} + \frac{\langle E|H|S\rangle\langle S|H|G\rangle}{\hbar(\omega_0 - \omega - \Lambda)}$. At resonance $\omega = \omega_0$, one can readily see from the matrix elements of H that $\Omega_{EG}^{\text{eff}}$ vanishes at the two fixed spot positions where $\Omega_2 = \Omega_1 e^{\pm \frac{i\pi}{2}}$. Overall, besides valuable information collected on the geometric configuration of the coupled emitters, the structuration of the laser field in amplitude and phase enables the inception of methods to

selectively address quantum-entangled states and build quantum interference pathways among them.

## Discussion

In summary, we demonstrate the manipulation of the degree of superposition in the superradiant and subradiant states of coherently coupled solid-state quantum emitters, using Stark shifts of their electronic levels. Nearly maximal superposition is achieved for emitters separated by distances up to several tens of nanometers in various dipole configurations. Moreover, a simple far-field optical method is developed to selectively excite the super- and subradiant states, which opens opportunities in quantum information schemes, in particular those exploiting the long-lived radiative decay of subradiant states. Spatial signatures inherent to dipole–dipole interaction are unveiled with ESSat nanoscopy and provide a unique mean to localize the coupled emitters. This far-field versatile method could be extended to engineer the spectroscopic selection rules for various molecular or nanometer scale systems. The present study opens up the opportunity to perform fast manipulation of entanglement for quantum logic gate operations[53], using rapid entanglement-disentanglement operations on molecular qubits with short electric field pulses. It also enables investigations of the rich quantum signatures of the light emitted from collective delocalized excitations, such as the $|E\rangle \rightarrow |S\rangle \rightarrow |G\rangle$ cascade that can be used to generate time-energy entangled photon-pairs[54,55].

Future investigation of the entanglement in systems scaled up to arrays of two-level emitters should focus on highly directional scattering properties[24], highly nonlinear responses with few photons[56] and should set the foundations for platforms of light–matter interfaces. Scaling up is challenging since currently fluorescent molecules are randomly distributed in their solid hosts. Chemical synthesis methods of molecular dimers with controlled separation distance are emerging[57] and could be extended to multimers of interacting molecules. Other routes consist in using encapsulation of fluorescent molecules in nanotubes[58–60], or nanoprinting methods enabling the deposition of subwavelength-sized crystals hosting a countable number of photostable and oriented molecules with subwavelength positioning accuracy[61].

The experimental and theoretical tools developed here to study test-bench entangled molecular pairs should also foster thorough investigations of a wealth of elaborate coupled systems, such as polymer conjugates[62] and quantum dot molecules[9]. It will be particularly interesting to explore the decoherence processes in highly delocalized molecular systems. For instance, light-harvesting complexes[63] are composed of a very dense set of fluorophores separated by few nanometers, leading to extremely strong dipolar coupling between molecules and multipartite quantum entanglement that survives over picosecond timescales at room temperature despite decoherence effects associated with the surrounding phonon bath[64–66]. The robustness of these quantum systems against decoherence mechanisms are still poorly understood. Open questions concern the coupling of such delocalized states to phonons[67,68] and the possible existence of non-Markovian effects. With the aim of deciphering the decoherence mechanisms in these complex systems, a pair of coupled molecules can therefore be the ideal test-bench elementary brick, starting with the temperature dependence of its degree of entanglement.

## Methods

**Sample preparation**. A small piece of naphthalene doped with DBATT fluorophores is heated above the melting point (~ 80 °C). A drop of this solution with DBATT concentration ~ $10^{-6}$ M is then pressed between this coverslip equipped with electrodes and an ordinary glass coverslip. After cooling down to room temperature, a thin polycrystalline layer with a thickness less than 5 μm is obtained.

The upper glass coverslip is removed and replaced by a spin-coated layer of polyvinyl alcohol (with thickness less than a few hundreds of nm) to avoid naphthalene evaporation during the sample assembly on the 2D piezo-scanning stage and the cooling process down to liquid helium temperature.

**Stark tuning of the molecular resonance frequencies**. Under the application of a static electric field **E**, single DBATT molecules embedded in a naphthalene crystal often exhibit both a linear and a quadratic Stark shift $h\delta\nu$ of their optical resonance, which are connected to the changes in static dipole moment $\delta\mu$ and in polarisability tensor $\delta\alpha$ between their ground and excited electronic states[42]:

$h\delta\nu = -\delta\mu.\mathbf{E} - \mathbf{E}\delta\alpha\mathbf{E}/2$. The quadratic contribution related to $\delta\alpha$ (set by the molecular volume) is very similar among molecules at the voltages used in this study. The differential Stark shift of two DBATT molecules, which is the relevant parameter for tailoring the degree entanglement within their $|S\rangle$ and $|A\rangle$ states, is thus essentially set by their difference in $\delta\mu$ along the applied field. Distortions of DBATT molecules in their insertion site indeed break their centrosymmetry, leading to a residual dipole moment up to few milliDebyes. The associated linear Stark coefficient is usually less than 10 MHz/(MV m$^{-1}$).

**Determination of the molecular and laser interaction parameters**. Modeling the experimental spectra measured for a pair of coupled molecules under a Gaussian-shaped or $LG_{01}$-shaped laser beam requires the knowledge of the dipole orientations and the locations of the molecules. We explain here the method used to extract these parameters, taking the example of the coupled molecular pair presented in Figs. 1b and 4.

As exemplified in Supplementary Fig. 8, we reproduce a stack of excitation spectra recorded at zero-voltage for different laser intensities with the following fitting parameters: The radiative decay rate $\gamma_0/2\pi = 21.5$ MHz, the interaction strength $V = -17\,\gamma_0$, the frequency mismatch $\Delta = 10\gamma_0$ between molecules, the cross-damping rate $\gamma_{12} = 0.3\gamma_0$, the saturation intensity $I_{sat} = 40$ W cm$^{-2}$, which is the same for both molecules, in consistence with a configuration of dipoles with nearly parallel orientations.

In order to stress the robustness of this set of parameters, it is worth noting that:

(i)　The value of the interaction strength $V = -17\gamma_0$ is supported by the Stark-tuned spectral trails of Supplementary Fig. 4, which display a waist of ~ $2V$ between the superradiant and subradiant lines.

(ii)　The nearly parallel-dipole configuration ($\Omega_1 = \Omega_2$) is supported by the vanishing fluorescence intensity stemming from the subradiant state in Supplementary Fig. 4 and by the determination of the dipole projections in the focal plane, given in Supplementary Fig. 9.

(iii)　The cross-damping rate is consistent with the expression $\gamma_{12} = \alpha\gamma_0\left(\hat{\mathbf{d}}_1.\hat{\mathbf{d}}_2\right)$ derived when the distance between molecules is much less than the optical wavelength, taking $\alpha = 0.3$ and parallel dipoles[44].

(iv)　The polarization diagrams of the fluorescence emission provide the orientations of the dipole projections in the focal plane. From Supplementary Fig. 9, we deduce that the in-plane angular separation between the two dipoles is less than 5°.

The in-plane distance between the emitters is extracted from the amplitude ratio of the ZPLs associated to the allowed transitions $|G\rangle \rightarrow |S\rangle$ (Gaussian beam illumination) and $|G\rangle \rightarrow |A\rangle$ (doughnut beam illumination), when the spectra are recorded under the same excitation power. As stressed in Fig. 4 and Supplementary Fig. 11, different locations of parallel dipoles in various planes of the sample produce the same spectral fingerprints. To reproduce the experimental spectra of Fig. 4, we have used $\mathbf{r}_1 = [-9, 0, 0]$ nm, $\mathbf{r}_2 = [9, 0, -13.4]$ nm, and dipole orientations $\hat{\mathbf{d}}_1=[1, 0, 0]$, $\hat{\mathbf{d}}_2 = [1, 0, 0]$ for the two molecules.

## Data availability

All data that support the conclusions of this study are included in the article and the Supplementary Information file. These data are available from the corresponding author upon reasonable request.

## Code availability

Numerical codes used in this study are available from the corresponding author upon reasonable request.

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

## Acknowledgements

The authors acknowledge the financial support from the French National Agency for Research, Région Aquitaine, Institut Universitaire de France, Idex Bordeaux (LAPHIA Program), and the EUR Light S&T Graduate Program (PIA3 Program "Investment for the Future", ANR-17-EURE- 0027), and GPR LIGHT. We are grateful to E. Cormier for loaning a fast pulse generator, and D. Mailly (C2N) for help in the fabrication of the electrodes.

## Author contributions

J.-B.T., Q.D. performed the experiments and the theoretical simulations. J.-B. T., P.T., and B.L. wrote the manuscript. B.L. supervised the project.

## Competing interests

The authors declare no competing interests.
