## [Peer Review File · Nature Communications]

Tailoring the superradiant and subradiant nature of two coherently coupled quantum emittersREVIEWER COMMENTS

Reviewer #1 (Remarks to the Author):

In the manuscript entitled “Tailoring the degree of entanglement of two coherently coupled quantum emitters” the authors present their theoretical and experimental work on the energy entanglement between two PAH molecules, resulting from dipole-dipole coupling at low temperature. The candidate pairs are selected by means of hyperspectral imaging, while Stark shift allows to compensate for the residual inhomogeneous broadening, thus maximizing delocalized excitation. Evidence of dipole-dipole coupling is found by looking at two-photon absorption in the excitation spectrum as a function of pump power.

The degree of entanglement is estimated from the linewidths of the splitted super- and sub-radiant states. The set of experimental results is rich, comprises also auto-correlation measurements and is thoroughly backed up by a theoretical model, that takes into account the excitation beam spatial dependence. Selective preparation of maximally entangled states is indeed demonstrated by using different Laguerre-Gaussian laser modes in excitation.

The novelty of this papers is in the details of the experiments and in particular in the combination of different high-resolution imaging techniques and low-temperature molecular spectroscopy. The manuscript is well organized and clear. In my view, the impact of the findings could be better presented. Provided my comments below are satisfactorily addressed, I would definitely recommend this paper for publication.

1. The application of this system for quantum information processing is not clear to me (how long do these entangled states live with respect to the time needed to perform a logic operation? How does this compare to other known platforms for quantum information processing?). Neither it is obvious how these results could be extended to understand the coherent coupling in biological systems. The authors should be either more precise or refine their claims both in the abstract, in the introduction and in the conclusions.

2. Entanglement fidelity should be calculated and the relative error estimated, to allow for comparison with other systems, that would be definitely relevant for a Nature Comm. paper. Moreover, as in the paper no entanglement witness is mentioned or plotted as a function of some parameter, I would strongly encourage a revision of the title.

3. I think that literature is not always properly cited. For instance, Refs 6 and 7 do not refer to the integration of quantum emitters as one would expect. Maybe Ref 6 could go together with 5? More recent works about integrating emitters in photonic devices for quantum technologies could be cited, see for molecules e.g. Nat. Mater. (2021). <https://doi.org/10.1038/s41563-021-00987-4>, Advanced

quantum Technologies (2019) <https://doi.org/10.1002/qute.201900020> for quantum dots or analogous reviews for diamond color centers.

4. It would be interesting to estimate the super and subradiant decay rates from lifetime measurements. Moreover, it would be instructive to see a plot of such rates as a function of the detuning.

5. The authors claim that maximal entanglement can be achieved even with emitters relatively far apart. Could they explain in the text more explicitly how r_{12} is obtained from the measurements?

6. I think one important aspect that is only marginally addressed in the discussion is the properties of the emitted light. Besides bunching and anti-bunching, is there anything peculiar or potentially relevant about the generated state of light? One interesting reference in this respect could be "Nanoscale continuous quantum light sources based on driven dipole emitter arrays Appl. Phys. Lett. 119, 024002 (2021); <https://doi.org/10.1063/5.0049270>"

7. Along the same line, I wonder how doable it would be to scale up the interaction to a larger number of molecules. That would create a totally different paradigm. Surprisingly the authors do not discuss this perspective.

Costanza Toninelli

Reviewer #2 (Remarks to the Author):

In order to control and manipulate quantum-entangled non-local states, this paper attempts a quite novel and sophisticated technique, and reports interesting findings. Implementing hyperspectral imaging, it reaches distinctive spectral signatures of molecular entanglement. With a laser field tailored in amplitude and phase, entangled delocalized states can be selectively prepared. It suggests a test-bench to decipher complex physical mechanisms and paves the way towards practical quantum information processing. So, this paper is almost acceptable for Nature Communication. However, some doubts should be cleared out before final decision.

1. It reports coherent coupling and even entanglement between two chemical molecules, The molecule size may be on the order of nanometer, then its electronic state wavefunctions, including ground and excited, should be difficult to extend out of the molecule for more than a few nm. If such two molecules are separated by 60 nm, their electronic states can hardly overlap and interact with each other. So, it is better to physically explain why the strong coupling over distance much longer than the molecule size is possible. Is it that the dipole interaction range can be much larger than the geometric size of the dipoles?

2. External electric field is applied to the molecules to produce Stark effect. Increasing electric field might change the charge distribution of the molecule, and thus change the molecular dipole itself in

orientation, strength etc. It is better to have an analysis about whether dipole variation due to external electric field is too small to be considered in characterizing dipole coupling.

3. The entanglement in mid-states $|S\rangle$ and $|A\rangle$ can be so strong. Then, the “biexciton” state $|E\rangle$ can also be a highly entangled state. The transition of this state $|E\rangle$ to the ground $|G\rangle$ will give an entangled photon pair, especially in the cases of nearly degenerating $|S\rangle$ and $|A\rangle$. Probably, this pairs of entangled photons can more effectively show the good and distant dipole coupling, generation and manipulation of molecular entanglement. Then, is there any job to get an entangled photon pair associated with $|E\rangle$? If this photon entanglement is hardly realizable, the claimed entanglement of $|S\rangle$ and $|A\rangle$ states may be questionable.

Reviewer #3 (Remarks to the Author):

With hyperspectral fluorescence microscopy, the authors identify molecular pairs that are close to each other spatially. These molecular pairs couple to each other via dipole-dipole interactions. The resonance frequencies are adjusted through Stark shift with a static electric field. The emergence of the central resonances upon an increase in excitation intensities as well as tuning closer to resonance signify a two-photon simultaneous excitations of both molecules. It also serves as an evidence of the presence of dipole-dipole interaction.

With these techniques, they also identify a superradiant state and a subradiant state, signified by a broadening/narrowing of the linewidth in the fluorescence spectra. Moreover, with some careful beam shaping, they can selectively excite either the superradiant or the subradiant state. Photon antibunching and bunching are observed when they excite the two-molecule system to a single state (superradiant and subradiant state), and the double-excited state via two photon excitations, respectively. Finally, they demonstrate a novel nanospectroscopy technique to determine the locations of the two emitters with a nanometric precision.

The two Bell states in this work have superradiant and subradiant features. However, to unambiguously demonstrate entanglement, a direct measurement of the off-diagonal terms in the two-qubit density matrix is required. A major shortcoming of this work is that the authors fail to show such direct evidence of entanglement. The authors could do this measurement via state tomography or parity oscillations as shown in [Phys. Rev. A 82, 030306(R), Phys. Rev. Lett. 81, 3631–3634 (1998)]. These measurements require performing single qubit rotation as well as a joint measurement of the two-qubit state. Selective single qubit rotation is on its own an important ingredient for a quantum information system, and is not demonstrated in the present work.

On top of this, the author suggests that the system is designed with quantum information processing in mind. With this suggestion, the readers would want to see more information about the system towards quantum information processing applications, including, qubit state preparation and readout, single qubit coherence time, single qubit gate fidelity, two-qubit gate fidelity, Bell state coherence time, and the scalability of the system and control, etc.

For example, the authors do not provide information on how these states could be used for further quantum entanglement applications. In particular, for most applications, one needs to be able to initialize and prepare the molecules in given states in a controllable fashion. However, it is unclear how one can coherently drive from $|gg\rangle$ to one of $|S\rangle$ or $|A\rangle$ deterministically in the present system. Nor is it shown how one can detect $|S\rangle$ or $|A\rangle$. Furthermore, the coherence times demonstrated here are too short compared to the oscillation times rendering them unusable for most applications.

Therefore, given the identified shortcomings, I do not recommend the paper for publication in Nature Communication at the current stage.

In addition to these high-level concerns above, some further comments are listed below:

Figure 1: Fig 1b is difficult to understand without axis labels. The caption mentions a green arrow in Fig. 1 c/e, but none is given in the figure.

It would help if the authors could provide some intuition as to why the symmetry of the ZPL's are different for Figs 1d-f depending on dipole configuration.

Figure 2 is confusing to follow with the multitude of plots with some overlapping information.

Fig 2 c/d - how are the two resolved? Does this use Gaussian vs Doughnut beam to resolve the two peaks?

The inset of Fig 2b and Fig 2f are qualitatively inconsistent (the "v" opens towards higher voltages in 2b, while it is the opposite for Fig 2f). In addition, where is the two-photon transition in the inset of Fig 2b?

It is unclear whether the electric field applied is constant or a gradient across the emitters to tune the relative frequency difference.

Is there a calibration for the Stark shift from voltage to frequency units?

In Fig 2e the linewidth is plot as a function of excitation power and extrapolated to zero power. Have the authors considered time broadening and other potential broadening mechanisms?

Figure 4: The scale for the inset of Fig 4a is unclear

What is the coherence time limited by? Is it fundamental due to the matrix environment?

The Rabi oscillations between $|G\rangle$ and $|A\rangle$, and $|G\rangle$ and $|S\rangle$ are demonstrated through coincidence measurement. Is there a way to directly measure the Rabi oscillation? For example, by monitoring the fluorescence as a function of time.

Color code:

Blue for the Reviewers' remarks;

Black for the Authors' response;

Green for the modifications in the manuscript or Supplementary Information.

Response to Reviewer# 1

We thank the Reviewer for positive comments on our work, and recommendation of this paper for publication provided that the Reviewer's comments are satisfactorily addressed. We develop below the point-to-point answers to the questions and comments of the Reviewer.

1. The application of this system for quantum information processing is not clear to me (how long do these entangled states live with respect to the time needed to perform a logic operation?)

At liquid helium temperatures there is no thermal dephasing in these molecular systems. The coherence time and therefore the lifetimes of the states $|S\rangle$ and $|A\rangle$ can readily be extracted from the widths of the ZPLs in the fluorescence excitation spectra (Fig 2). The lifetimes depend on the degree of entanglement in $|S\rangle$ and $|A\rangle$ tailored by Stark effect (see response to point #4) and those of the corresponding ideal Bell states are expected to be $[\gamma_0(1 \pm \alpha)]^{-1}$. Depending on α , the lifetime lies between $(2\gamma_0)^{-1}$ and γ_0^{-1} for the symmetric state, while it is longer for the antisymmetric state and extendable when tailoring α close to 1 [Wang et al Nature Physics 15 (2019) 483] or choosing a system displaying intrinsically a high α , such as silicon-vacancy centers in diamond [Lindner et al. New J Phys. 20 (2018) 115002].

To compare the coherence time with the time of a quantum logic gate operation, we have studied the response of a single molecule and a coupled pair of molecules to pulses resonant with the $|g\rangle \rightarrow |e\rangle$ transition and with the $|G\rangle \rightarrow |S\rangle$ transition, respectively (See appended figure). Fast and highly contrasted Rabi oscillations are observed in the fluorescence of a single molecule submitted to a transient resonant excitation (Panel a). Using a laser pulse with suited duration, efficient preparation of the excited state can be achieved within times ~ 2 ns and with a probability of 0.9 corresponding to the π -pulse fidelity. This time can even be shortened and the fidelity improved by using pulses with faster rise times and larger peak intensities. Panel b (given for the Reviewer only) shows preliminary measurements of Rabi oscillations observed under pulsed excitation resonant on the $|G\rangle \rightarrow |S\rangle$ transition of a pair of coupled molecules. It shows the possibility of fast preparation of the system in $|S\rangle$ starting from the ground state $|G\rangle$. Fast entanglement/disentanglement operations on the two qubits could be achieved using resonant ns laser pulses and short electric field pulses (Stark shifts), along similar lines to what has been done with superconducting qubits [see for example, Steffen et al. Science 313 (2006) 1423]. Such preliminary

results on entanglement manipulation together with the realization of quantum logic gates require more experimental efforts and developments: They are beyond the scope of this manuscript and will be the subject of another study.

Single-qubit and two-qubit rotation.

a, Temporal evolution of the fluorescence signal from a single DBATT molecule upon pulsed resonant excitation (Integration time 25 min, bin width 64 ps). The fluorescence background stemming from out-of-focus molecules has been subtracted and the fluorescence signal is normalized to 1/2 at long times in order to mimic the evolution of the excited state population. The black curve is a theoretical simulation of the excited state population, which reproduces the damped Rabi oscillations following the rising edge of the excitation pulse, with a Rabi angular frequency $11.7 \gamma_0$ and an optical coherence lifetime $T_2 = 2T_1$. The excited state lifetime $T_1 = (7.39 \pm 0.02)$ ns is deduced from the exponential decay following the falling edge of the pulse. **b**, Rabi oscillations under pulsed excitation resonant on the $|G\rangle \rightarrow |S\rangle$ transition of a pair of coupled molecules. The black dotted line corresponds to the laser pulse.

The panel a of the appended figure and a scheme of the experimental setup have been incorporated as Supplementary Fig. 1 of the revised manuscript, together with the following sentences in the main text:

“Such molecules nearly behave like simple two-level systems (...) allowing quantum optical measurements at strong excitation intensities³². For instance, Rabi oscillations in the fluorescence of a single DBATT molecule submitted to a transient resonant excitation demonstrate the possibility to prepare the excited state with a probability 0.9 after a π -pulse and show that the coherence dephasing time reaches its upper limit given by twice the excited state lifetime (Supplementary Fig. 1).”

How does this compare to other known platforms for quantum information processing?).

We thank the Reviewer for raising this question, which will help us clarifying the main goals of our manuscript. An obvious misunderstanding comes from the awkward expression “quantum information processing” used twice in the abstract,

while we did not address the realization of a quantum logic operation. We have taken this opportunity to rephrase the abstract and compact it to comply with the editorial guidelines. The abstract now reads:

“The control and manipulation of quantum-entangled states is crucial for the development of quantum technologies. A promising route is to couple solid-state quantum emitters through their optical dipole-dipole interactions. Entanglement in itself is challenging, as it requires both nanometric distances between emitters and nearly degenerate electronic transitions. Implementing hyperspectral imaging to identify pairs of coupled molecules, we reach distinctive spectral signatures of maximal entanglement of the superradiant and subradiant electronic states by tuning the molecular optical resonances with Stark effect. We demonstrate far-field selective excitation of the long-lived subradiant delocalized states with a laser field tailored in amplitude and phase. Optical nanoscopy of the entangled molecules unveils novel spatial signatures that result from quantum interferences in their excitation pathways and reveal the location of each emitter. Controlled molecular entanglement will help deciphering more complex physical or biological mechanisms governed by the coherent coupling and developing new quantum information schemes.”

Nevertheless, we now address the point raised by the Reviewer. Many experimental platforms are currently being investigated for building a large-scale quantum computer (e.g. photon systems, neutral atoms, trapped-ions, nitrogen vacancy centers, quantum dots, superconducting circuits...). For instance, platforms based on superconducting qubits [Kjaergaard et al *Ann. Rev. Cond. Mat. Phys.* 11 (2020) 369], trapped ions [Monroe et al. *Rev. Mod. Phys.* 93 (2021) 025001] or Rydberg atoms [Scholl et al. *Nature* 595 (2021) 233] have reached after long term and intense research high two-qubit gate fidelities together with long coherence times. Each of these physical platforms offers different advantages as well as drawbacks. Interfacing and integrating different platforms could exploit unique advantages of each platform in a single setting where different quantum processing tasks would be delegated to the best suited system. Hybrid quantum systems combining in a well-controlled manner different quantum states from the fields of optics, atomic physics and condensed matter physics could indeed achieve capabilities that are not available with either system alone. Studying new quantum systems for hybridization can be key to coherently control complex quantum properties needed for novel quantum-enhanced technologies. The development of molecular systems for control and manipulation of quantum-entangled non-local states is in its infancy. Molecular systems present several advantages:

- Molecules are readily trapped in a solid matrix are simple to handle for continuous measurements over days. They require moderately low temperatures, while extremely low temperatures are needed for superconducting qubits. Complex methods of trapping and cooling of atomic systems are heavy to implement and may need reloading procedures.

- Another advantage of molecular systems lies in the fact that the separation distance between molecules can be as short as few nanometers, while distance between trapped atoms or ions is in the order of few micrometers. As a consequence, the coupling constant of molecular systems exceeds that of atomic systems and reach the GHz range, enabling fast (sub ns) manipulation of the entanglement and quantum logic gates operations. Couplings are also larger than those achieved with superconducting qubits (typically below 100 MHz).
- It could be possible to reach a high density of qubits with molecular systems. Currently, the molecules are randomly located in the matrix, but one can rely on chemistry developments to produce macromolecules with a controllable number of molecules and positioning geometries [Hübner et al. PRL 91 (2003) 093903; Feofanov et al. ChemistrySelect 6 (2021) 10671] (see point#7).

Neither it is obvious how these results could be extended to understand the coherent coupling in biological systems. The authors should be either more precise or refine their claims both in the abstract, in the introduction and in the conclusions.

The revised abstract mentions that “Controlled molecular entanglement will help deciphering more complex physical or biological mechanisms governed by the coherent coupling...”. Moreover, in order to make more explicit the interest of studying molecular entanglement for the comprehension of biological systems, we have added the following discussion in the conclusion:

“It will be particularly interesting to explore the decoherence processes in highly delocalized molecular systems. For instance, light-harvesting complexes [Engel et al. Nature 446 (2007) 782] are composed of a very dense set of fluorophores separated by few nanometers, leading to extremely strong dipolar coupling between molecules and multipartite quantum entanglement that survives over picosecond timescales at room temperature despite decoherence effects associated with the surrounding phonon bath [Sarovar et al. Nature Phys 6 (2010) 462; Collini et al. Nature 463 (2010) 644 ; Chin et al. Nature Phys 9 (2013) 113]. The robustness of these quantum systems against decoherence mechanisms are still poorly understood. Open questions concern the coupling of such delocalized states to phonons and the possible existence of non-Markovian effects. A pair of coupled molecules can therefore be the ideal elementary brick to help deciphering the decoherence mechanisms in these complex systems.”

2. Entanglement fidelity should be calculated and the relative error estimated, to allow for comparison with other systems, that would be definitely relevant for a Nature Comm. paper. Moreover, as in the paper no entanglement witness is mentioned or plotted as a function of some parameter, I would strongly encourage a revision of the title.

In this paper we do not aim nor claim time-manipulation of the entanglement of two coherently coupled single molecules. We instead aim at demonstrating for the first

time that one can tailor the degree of entanglement (as defined in Ref. [Abouraddy et al PRA 64 (2001) 050101(R)]) in the superradiant and subradiant states that are commonly considered as entangled symmetric and antisymmetric states [see for example Ficek & Tanas Physics Reports 372 (2002) 369; Hebenstreit et al. PRL 118 (2017) 143602]. We may have been awkward in the caption of Fig. 2 and a few sentences of the manuscript, by using the expression “*manipulation of entanglement*” instead of referring to the manipulation of the degree of entanglement in the superradiant and subradiant states $|S\rangle$ and $|A\rangle$. To avoid any confusion, we have brought this precision in the title, which now writes “Tailoring the degree of entanglement in the superradiant and subradiant states of two coherently coupled quantum emitters” and replaced “manipulation of entanglement” by “manipulation of the degree of entanglement in the states...” in the text. The abstract also puts more emphasis on reaching “maximal entanglement of the superradiant and subradiant electronic states”.

Therefore, rather than using the two-qubit state preparation fidelity, we find more relevant to use in the manuscript the notion of degree of entanglement in the pure states $|S\rangle$ or $|A\rangle$, as introduced in [Abouraddy et al. PRA 64 (2001) 050101(R)]. In the revised version of the manuscript, we have added the following sentences:

“As defined in Ref. [Abouraddy et al. PRA 64 (2001) 050101(R)] for pure bipartite two-qubit states, the degree of entanglement P_E of the two pure states $|S\rangle$ or $|A\rangle$ is obtained from their decomposition $p|\Psi_{Bell}\rangle + \sqrt{1-p^2}e^{i\phi}|\Psi_{fact}\rangle$ into two orthogonal quantum states, $|\Psi_{Bell}\rangle$ being the corresponding Bell state, $|\Psi_{fact}\rangle$ a factorizable state, p and ϕ real numbers. The degree of entanglement is defined by $P_E = p^2 = 2|ab|$ and is identical to the concurrence [Hill & Wootters, Phys. Rev. Lett. 78 (1997) 5022]. For the pair of molecules presented in Fig. 2f, P_E reaches $(95 \pm 5)\%$ at the maximal Stark voltage. Interestingly, this value sets an upper bound to the fidelity achievable when optically preparing the system in the $|S\rangle$ or $|A\rangle$ states from the ground state $|G\rangle$, using a resonant π -pulse excitation.”

Incidentally, throughout the manuscript we have provided several signatures of entanglement in the states $|S\rangle$ and $|A\rangle$ and compared the situations of high and low entanglement regimes of these states via Stark shifts of the molecular optical resonances. A further comparison is now added to the revised manuscript. The appended figure shows that the Rabi oscillations in the autocorrelation function are different when exciting the states $|S\rangle$ or $|A\rangle$. Indeed, panels a,b show evidence for a pronounced difference in the Rabi frequencies when the degree of entanglement is maximal, while Rabi frequencies become similar when lowering the degree of entanglement by Stark shifting the molecular resonances apart (Panels c,d). These panels and caption have been inserted in the revised manuscript, together with rephrased sentences picked up in this paragraph (see list of changes).

a,b, Normalized photon coincidence histograms (after background subtraction) measured for a pair of coupled molecules excited with a Gaussian-shaped laser beam of intensity 200 W cm^{-2} at resonance with the $|G\rangle \rightarrow |S\rangle$ transition (**a**) and with the $|G\rangle \rightarrow |A\rangle$ transition (**b**) when the degree of entanglement is maximized by Stark effect ($P_E \sim 0.92$, voltage 150 V). They are characterized by an antibunching dip and damped Rabi oscillations, while photon bunching is evidenced when the laser is resonant on the two-photon transition (Supplementary Fig. 10). The second-order correlation functions (red curves) are computed for nearly parallel dipoles in the J-configuration with $V = -17 \gamma_0$, $\gamma_{12} = 0.3 \gamma_0$, and $\Delta = 10 \gamma_0$ (Supplementary Note 1). **c,d**, Photon coincidence histograms obtained for the same pair of coupled molecules when the molecular resonances are tuned apart by Stark effect (-150 V), leading to a modest degree of entanglement $P_E \sim 0.24$. The excitation intensity is 40 W cm^{-2} . The simulations are performed with the same parameters except for $\Delta = 60 \gamma_0$.

3. I think that literature is not always properly cited. For instance, Refs 6 and 7 do not refer to the integration of quantum emitters as one would expect. Maybe Ref 6 could go together with 5? More recent works about integrating emitters in photonic devices for quantum technologies could be cited, see for molecules e.g. Nat. Mater. (2021). <https://doi.org/10.1038/s41563-021-00987-4>, Advanced quantum Technologies (2019) <https://doi.org/10.1002/qute.201900020> for quantum dots or analogous reviews for diamond color centers.

We thank the Reviewer for this comment. We have grouped Refs. 5 and 6. and have replaced 7 by these two references in the revised manuscript.

4. It would be interesting to estimate the super and subradiant decay rates from lifetime measurements. Moreover, it would be instructive to see a plot of such rates as a function of the detuning.

As requested by the Reviewer, we have supplemented our experimental study with direct measurements of $|S\rangle$ and $|A\rangle$ lifetimes and their evolutions as the degree of entanglement is varied by Stark effect. These data are presented in the appended figure and are the subject of a new figure (Fig. 3) and caption in the manuscript, together with the following text:

“The super- and sub-radiance character of the $|S\rangle$ and $|A\rangle$ states is further evidenced by measurements of their lifetimes for various degrees of entanglement. Figure 3a displays the normalized PL decay curves recorded after selective, pulsed excitation of the $|S\rangle$ state (blue curve) and $|A\rangle$ state (red curve) in the situation where the molecular detuning Δ is minimized by Stark effect. These decays are well reproduced by single exponential curves with lifetimes 6.3 ns and 11.1 ns, respectively, which are markedly different from the average lifetime 7.8 ± 0.34 ns of single molecules found in this sample, as shown in the inset of Fig. 3a. One can notice that the average decay rate $(\gamma_+ + \gamma_-)/2$ coincides with the decay rate of the uncoupled single molecules, as a signature of transfer of oscillatory strength between the subradiant and superradiant states. Moreover, we find a good agreement of these lifetimes with the computed values $(\gamma_0 \mp 2ab\gamma_{12})^{-1}$ plotted as red and blue curves in Fig. 3b.”

Fig 3 Comparison of the subradiant and superradiant lifetimes.

a, Decays curves of the subradiant and superradiant states. The laser pulses have a rise-fall time of 1 ns and a repetition rate 100 kHz. The solid curves are exponential fits with lifetimes $\tau_+ = 6.3$ ns and $\tau_- = 11.1$ ns, which are consistent with the computed subradiant lifetime $\tau_- = (\gamma_0 - 2ab\gamma_{12})^{-1} = 1.43 \gamma_0^{-1}$ and the superradiant one $\tau_+ = (\gamma_0 + 2ab\gamma_{12})^{-1} = 0.77 \gamma_0^{-1}$, using $a = 0.88$, $b = 0.47$, $\gamma_{12} = 0.35 \gamma_0$. The coefficients a and b are deduced from the diagonalization of the Hamiltonian H in the absence of laser field. The detuning $\Delta = 15 \gamma_0$ is derived from the splitting between $|S\rangle$ and $|A\rangle$ and from the coupling constant $V = 9.5 \gamma_0$ that is deduced from the fits of the excitation spectra at various excitation intensities. Inset: Histogram of the lifetime of 35 uncoupled single molecules. The blue and red bars indicate the values of τ_+ and τ_- for the coupled pair. **b**, Evolution of τ_+ (blue circles) and τ_- (red circles) with the molecular detuning Δ which is varied by differential Stark shifts of the molecular resonances from $15 \gamma_0$ (150 Volts) to $32 \gamma_0$ (0 Volt). The solid curves are the computed values of τ_+ and τ_- . The black circles are the inverse of the average subradiant and superradiant decay rates (τ_+^{-1} and τ_-^{-1}), and coincide with the average lifetime of the uncoupled single molecules (black dashed line).

5. The authors claim that maximal entanglement can be achieved even with emitters relatively far apart. Could they explain in the text more explicitly how r_{12} is obtained from the measurements?

For the pair of molecules far apart (~ 60 nm separation), the ZPL width of the subradiant state is 13 MHz when the degree of entanglement is maximal ($2ab \sim 1$). Such narrow linewidth points to a subradiant decay rate $\gamma_- = \gamma_0(1 - 2ab\alpha \hat{d}_1 \cdot \hat{d}_2)$ reaching its lower bound $\gamma_0(1 - \alpha) \sim 0.65\gamma_0$ obtained when the transition dipole moments of the molecules are parallel. Knowing that the molecules are in the J dipole configuration from the excitation spectrum (blue-shifted ZPL of the $|A\rangle$ state), one can directly extract the separation distance from the coupling V measured from the spectral separation ($2V$) of the $|S\rangle$ and $|A\rangle$ ZPLs at maximal entanglement (anticrossing point), i.e. using $V = -3\alpha\gamma_0/2(kr_{12})^3$.

In the revised manuscript, we have added the information that the spectral separation of the $|S\rangle$ and $|A\rangle$ ZPLs is $2V$ at the minimal detuning $|\Delta|$, and added the expression $V = -3\alpha\gamma_0/2(kr_{12})^3$ to justify the separation distance (see list of changes).

6. I think one important aspect that is only marginally addressed in the discussion is the properties of the emitted light. Besides bunching and anti-bunching, is there anything peculiar or potentially relevant about the generated state of light? One interesting reference in this respect could be “Nanoscale continuous quantum light sources based on driven dipole emitter arrays Appl. Phys. Lett. 119, 024002 (2021); <https://doi.org/10.1063/5.0049270>”

We thank the Reviewer for mentioning this point that we now discuss in the revised conclusion (see point #7).

7. Along the same line, I wonder how doable it would be to scale up the interaction to a larger number of molecules. That would create a totally different paradigm. Surprisingly the authors do not discuss this perspective.

Scaling up is indeed challenging since up to now fluorescent molecules are randomly distributed in their solid hosts. Chemical synthesis methods of molecular dimers with controlled separation distance are emerging [Feofanov et al. ChemistrySelect 6 (2021) 10671] and could be extended to clusters of interacting molecules. Other routes consist in using encapsulation of fluorescent molecules in nanotubes [Cambré et al. Nature Nanotech 10 (2015) 248; Allard et al. Adv. Mater. 32 (2020) 2001419; Gaufres et al. 8 (2014) 72], or nanoprinting methods enabling the deposition of subwavelength-sized crystals hosting a countable number of photostable and oriented molecules with subwavelength positioning accuracy [Hail et al. Nature Comm. 10 (2019) 1880]. The control of multiple coupled molecules entangled in their subradiant and superradiant states could be performed with inhomogeneous electric fields and spectral selection of the entangled states.

Taking this point and point #6 into consideration (as well as remarks from the Reviewer#3), the revised conclusion is now enriched as follows:

“The present study also opens up the opportunity to investigate the rich quantum signatures of the light emitted from collective delocalized excitations, such as the $|E\rangle \rightarrow |S\rangle \rightarrow |G\rangle$ cascade that can be used to generate time-energy entangled photon-pairs [Simon and Poizat, PRL 94 (2005) 030502; Jayakumar et al. Nature Comm 5 (2019) 4251].

Future investigation of the entanglement in systems scaled up to arrays of two-level emitters should focus on highly directional scattering properties [Rui et al. Nature 583 (2020) 369], highly nonlinear responses with few photons [Holzinger et al. Appl Phys Rev 119 (2021) 024002] and should set the foundations for novel platforms of light-matter interfaces. Scaling up is challenging since currently fluorescent molecules are randomly distributed in their solid hosts. Chemical synthesis methods of molecular dimers with controlled separation distance are emerging [Feofanov et al. ChemistrySelect 6 (2021) 10671] and could be extended to multimers of interacting molecules. Other routes consist in using encapsulation of fluorescent molecules in nanotubes [Cambré et al. Nature Nanotech 10 (2015) 248; Allard et al. Adv. Mater. 32 (2020) 2001419; Gaufres et al. 8 (2014) 72], or nanoprinting methods enabling the deposition of subwavelength-sized crystals hosting a countable number of photostable and oriented molecules with subwavelength positioning accuracy [Hail et al. Nature Comm. 10 (2019) 1880].”

Response to Reviewer# 2

We thank the Reviewer for positive comments on our work, and consideration that our paper “*is almost acceptable for Nature Communication*”. “*However, some doubts should be cleared out before final decision*”. We develop below point-to-point answers to the questions and comments of the Reviewer.

1. It reports coherent coupling and even entanglement between two chemical molecules, The molecule size may be on the order of nanometer, then its electronic state wavefunctions, including ground and excited, should be difficult to extend out of the molecule for more than a few nm. If such two molecules are separated by 60 nm, their electronic states can hardly overlap and interact with each other. So, it is better to physically explain why the strong coupling over distance much longer than the molecule size is possible. Is it that the dipole interaction range can be much larger than the geometric size of the dipoles?

We fully agree with the Reviewer that the electronic states of both molecules do not overlap and interact, and there is no tunneling process. The optical dipole-dipole interaction addressed here is mediated by the electromagnetic field modes of the vacuum state [Akram et al PRA 62 (2000) 013413]. In the revised version of the manuscript, we have added this precision: “The vacuum-induced coupling between two molecules...” when introducing the expression of V .

The excited level of a single quantum emitter coupled to the vacuum field experiences both a Lamb shift and a decay of population due to spontaneous emission. In the same vein, the interaction of the bare molecules with the multimode vacuum field generates an incoherent exchange of excitation between molecules such that one spontaneously emits photons which are then absorbed by the other (which manifests in the cross damping rate γ_{12}). The interaction of the bare systems with the vacuum field also leads to a coherent coupling V between them, whose final expression [Akram et al PRA 62 (2000) 013413] recalls the familiar interaction of two classical dipoles, that can be derived as follows.

Classically, a dipole $\mathbf{d}_1 = d_1 \hat{\mathbf{d}}_1$ positioned at \mathbf{r}_1 and oscillating at an optical frequency $\omega_1 = ck_1/n$ generates an electric field at the position $\mathbf{r} = r \hat{\mathbf{r}}$ given by [Jackson, Classical Electrodynamics, Willey]:

$$\mathbf{E}_1(\mathbf{r}) = \frac{1}{4\pi\epsilon_0} \left\{ \left[\frac{3(\mathbf{r} - \mathbf{r}_1) \left((\mathbf{r} - \mathbf{r}_1) \cdot \hat{\mathbf{d}}_1 \right)}{|\mathbf{r} - \mathbf{r}_1|^3} - \hat{\mathbf{d}}_1 \right] \left(\frac{1}{|\mathbf{r} - \mathbf{r}_1|^3} - \frac{ik_1}{|\mathbf{r} - \mathbf{r}_1|^2} \right) e^{ik_1|\mathbf{r} - \mathbf{r}_1|} \right. \\ \left. + k_1^2 \left[\frac{\left((\mathbf{r} - \mathbf{r}_1) \times \hat{\mathbf{d}}_1 \right) \times (\mathbf{r} - \mathbf{r}_1)}{|\mathbf{r} - \mathbf{r}_1|^3} \right] \frac{e^{ik_1|\mathbf{r} - \mathbf{r}_1|}}{|\mathbf{r} - \mathbf{r}_1|} \right\}$$

The energy of interaction with the second dipole \mathbf{d}_2 located at \mathbf{r}_2 is given by $W = -\mathbf{d}_2 \cdot \mathbf{E}_1(\mathbf{r}_2)$, whose real part leads to the expression of V given in the manuscript when

$kr_{12} \ll 1$ (where $k = (k_1 + k_2)/2$) and whose imaginary part leads to the cross-damping rate γ_{12} . Such interaction exists over distances much larger than the physical extension of the molecular electronic clouds. For instance, in the J-configuration the coupling strength is $V \sim 18 \gamma_0 \sim 0.4$ GHz for a separation distance $r_{12} = 20$ nm.

2. External electric field is applied to the molecules to produce Stark effect. Increasing electric field might change the charge distribution of the molecule, and thus change the molecular dipole itself in orientation, strength etc. It is better to have an analysis about whether dipole variation due to external electric field is too small to be considered in characterizing dipole coupling.

Besides its oscillating dipole under the application of a laser field (optical transition dipole of the order of ten Debyes), a molecule has a static distribution of electric charges leading to a static electric dipole moment μ and a polarisability α . Due to their intrinsic centro-symmetric chemical structures, aromatic molecules such as DBATT should have a vanishing static electric dipole moment. However, these molecules may be distorted in their insertion site, leading to a residual static dipole moment. Under the application of a static electric field \mathbf{E} , single DBATT molecules embedded in a naphthalene crystal often exhibit both a linear and a quadratic Stark shift $h\delta\nu$ of their optical resonance, which are connected to the changes in dipole moment $\delta\mu$ and in polarisability tensor $\delta\alpha$ between their ground and excited electronic states:

$$h\delta\nu = -\delta\mu \cdot \mathbf{E} - \mathbf{E} \delta\alpha \mathbf{E} / 2$$

For this host-guest system, the linear Stark coefficient is usually less than 10 MHz/(MV/m), corresponding to $\delta\mu$ of a few mD, while the quadratic Stark coefficient is generally found between -1.5 and -6 MHz/(MV/m)² [Brunel et al. J. Phys. Chem. A 103 (1999) 2429]. The largest Stark shifts achieved are less than 1 GHz for a field of 10 MV/m, which represents in relative value less than 10^{-6} of the transition frequency (485 THz). Therefore, the static electric fields applied in this work induce an extremely weak perturbation of the molecular electronic clouds.

We have added the following sentences to the caption of the corresponding Supplementary Figure (renamed Supplementary Fig. 2):

“Under the application of a static electric field \mathbf{E} , single DBATT molecules embedded in a naphthalene crystal often exhibit both a linear and a quadratic Stark shift $h\delta\nu$ of their optical resonance, which are connected to the changes in static dipole moment $\delta\mu$ and in polarisability tensor $\delta\alpha$ between their ground and excited electronic states [Brunel et al. J. Phys. Chem. A 103 (1999) 2429]:

$h\delta\nu = -\delta\mu \cdot \mathbf{E} - \mathbf{E} \delta\alpha \mathbf{E} / 2$. The quadratic contribution related to $\delta\alpha$ (set by the molecular volume) is very similar among molecules at the voltages used in this study. The differential Stark shift of two DBATT molecules, which is the relevant parameter for tailoring the degree entanglement within their $|S\rangle$ and $|A\rangle$ states, is thus essentially

set by their difference in $\delta\mu$ along the applied field. Distortions of DBATT molecules in their insertion site indeed break their centrosymmetry, leading to a residual dipole moment up to few milliDebyes. The associated linear Stark coefficient is usually less than 10 MHz/(MV m⁻¹).”

3. The entanglement in mid-states $|S\rangle$ and $|A\rangle$ can be so strong. Then, the “biexciton” state $|E\rangle$ can also be a highly entangled state. The transition of this state $|E\rangle$ to the ground $|G\rangle$ will give an entangled photon pair, especially in the cases of nearly degenerating $|S\rangle$ and $|A\rangle$. Probably, this pairs of entangled photons can more effectively show the good and distant dipole coupling, generation and manipulation of molecular entanglement. Then, is there any job to get an entangled photon pair associated with $|E\rangle$? If this photon entanglement is hardly realizable, the claimed entanglement of $|S\rangle$ and $|A\rangle$ states may be questionable.

First, let us note that $|E\rangle$ is actually not an entangled state since it is a pure product state $|e_1, e_2\rangle = |e_1\rangle \otimes |e_2\rangle$. Maybe the Reviewer meant that the $|E\rangle$ state, which can be prepared using a π -pulse with a laser resonant on the two-photon $|E\rangle \rightarrow |G\rangle$ transition, could be used to prepare entangled photon-pairs. Indeed, we agree with the Reviewer that quantum signatures of the emitted light are interesting to explore, in particular for photons emitted on the zero-phonon lines. Note that the properties of two-photon cascades from the $|E\rangle$ state will be very different from those of biexciton recombination in quantum dots, where indistinguishable paths in two-photon cascades can be generated if the exciton states are degenerate. In the case of coupled molecules, the $|S\rangle$ and $|A\rangle$ states cannot be degenerate since their minimal energy-splitting is twice the coupling strength V . Moreover, when maximal entanglement is achieved, the cascade $|E\rangle \rightarrow |A\rangle \rightarrow |G\rangle$ via emission of photons on ZPLs involving the $|A\rangle$ state will be forbidden due to the zero-oscillator strength of the antisymmetric state $|A\rangle$. However, the allowed $|E\rangle \rightarrow |S\rangle \rightarrow |G\rangle$ cascade can be used to generate time-energy entangled photon-pairs [Simon and Poizat, PRL 94 (2005) 030502; Jayakumar et al. Nature Comm 5 (2019) 4251].

We have added the following sentences in the revised conclusion.

“The present study also opens up the opportunity to investigate the rich quantum signatures of the light emitted from collective delocalized excitations, such as the $|E\rangle \rightarrow |S\rangle \rightarrow |G\rangle$ cascade that can be used to generate time-energy entangled photon-pairs [Simon and Poizat, PRL 94 (2005) 030502; Jayakumar et al. Nature Comm 5 (2019) 4251].”

Response to Reviewer# 3

We thank the Reviewer for positive comments on our work. However, the Reviewer “identified shortcomings” and “*does not recommend the paper for publication in Nature Communication at the current stage.* We develop below point-to-point answers to the questions and comments of the Reviewer.

1. The two Bell states in this work have superradiant and subradiant features. However, to unambiguously demonstrate entanglement, a direct measurement of the off-diagonal terms in the two-qubit density matrix is required. A major shortcoming of this work is that the authors fail to show such direct evidence of entanglement. The authors could do this measurement via state tomography or parity oscillations as shown in [Phys. Rev. A 82, 030306(R), Phys. Rev. Lett. 81, 3631–3634 (1998)]. These measurements require performing single qubit rotation as well as a joint measurement of the two-qubit state.

In this paper we do not aim nor claim time-manipulation of the entanglement of two coherently coupled single molecules. We instead aim at demonstrating for the first time that one can tailor the degree of entanglement (as defined in Ref. [Abouraddy et al PRA 64 (2001) 050101(R)]) in the superradiant and subradiant states that are commonly considered as entangled symmetric and antisymmetric states [see for example Ficek & Tanas Physics Reports 372 (2002) 369; Hebenstreit et al. PRL 118 (2017) 143602]. We may have been awkward in the caption of Fig. 2 and a few sentences of the manuscript, by using the expression “*manipulation of entanglement*” instead of referring to the manipulation of the degree of entanglement in the superradiant and subradiant states $|S\rangle$ and $|A\rangle$. To avoid any confusion, we have brought this precision in the title, which now writes “*Tailoring the degree of entanglement in the superradiant and subradiant states of two coherently coupled quantum emitters*” and replaced “*manipulation of entanglement*” by “*manipulation of the degree of entanglement in the states...*” in the text. The abstract also puts more emphasis on reaching “*maximal entanglement of the superradiant and subradiant electronic states*”.

Indeed, we provide several signatures of entanglement in the states $|S\rangle$ and $|A\rangle$ in the CW experiments performed throughout this paper, and compare the situations of high and low entanglement regimes of these states via Stark shifts of the molecular optical resonances:

i) When carrying a high degree of entanglement, the superradiant and subradiant states $|S\rangle$ and $|A\rangle$ have different coupling to the laser mode. This is evidenced in the evolution of the fluorescence excitation spectra when the molecular transitions are brought to resonance by Stark effect (the ZPL of $|S\rangle$ widens while that of $|A\rangle$ narrows as the molecular detuning is decreased). Moreover, the Rabi oscillations in the autocorrelation function are different when exciting the states $|S\rangle$ or $|A\rangle$. As proof, the

appended figure displays new photon coincidence histograms measured on the same pair of coupled molecules. It shows evidence for a pronounced difference in the Rabi frequencies when the degree of entanglement is maximal (Panels a,b), while Rabi frequencies become similar when lowering the degree of entanglement by Stark shifting the molecular resonances apart (Panels c,d). These panels and caption have been inserted in the revised manuscript, together with the following sentences.

“Striking differences in coupling of $|S\rangle$ and $|A\rangle$ to the laser field manifest in the fluorescence intensity autocorrelation function $g^{(2)}(\tau)$ recorded under resonant excitation of $|S\rangle$ and $|A\rangle$ at identical laser intensities. Indeed, besides strong photon antibunching on both transitions, the normalized coincidence histograms show Rabi oscillations that are much faster on the $|G\rangle \rightarrow |S\rangle$ transition (Fig. 4a) than on the $|G\rangle \rightarrow |A\rangle$ one (Fig. 4b) when the degree of entanglement in $|S\rangle$ and $|A\rangle$ is maximal. In contrast, the Rabi frequencies become similar when lowering the degree of entanglement by Stark shifting these molecular resonances apart (Fig. 4c,d).”

a,b, Normalized photon coincidence histograms (after background subtraction) measured for a pair of coupled molecules excited with a Gaussian-shaped laser beam of intensity 200 W cm^{-2} at resonance with the $|G\rangle \rightarrow |S\rangle$ transition (**a**) and with the $|G\rangle \rightarrow |A\rangle$ transition (**b**) when the degree of entanglement is maximized by Stark effect ($P_E \sim 0.92$, voltage 150 V). They are characterized by an antibunching dip and damped Rabi oscillations, while photon bunching is evidenced when the laser is resonant on the two-photon transition (Supplementary Fig. 10). The second-order correlation functions (red curves) are computed for nearly parallel dipoles in the J-configuration with $V = -17 \gamma_0$, $\gamma_{12} = 0.3 \gamma_0$, and $\Delta = 10 \gamma_0$ (Supplementary Note 1). **c,d**, Photon coincidence histograms obtained for the same pair of coupled molecules when the molecular resonances are tuned apart by Stark effect (-150 V), leading to a modest degree of entanglement $P_E \sim 0.24$. The excitation intensity is 40 W cm^{-2} . The simulations are performed with the same parameters except for $\Delta = 60 \gamma_0$.

ii) ESSat super-resolution imaging shows spatial structures that reflect quantum interference between the two excitation paths that bring the system from $|G\rangle$ to $|E\rangle$: $|G\rangle \rightarrow |A\rangle \rightarrow |E\rangle$ and $|G\rangle \rightarrow |S\rangle \rightarrow |E\rangle$.

iii) The states $|S\rangle$ and $|A\rangle$ have different coupling to the vacuum field fluctuations and therefore different radiative lifetimes. Indeed, we have supplemented our experimental study with direct measurements of $|S\rangle$ and $|A\rangle$ lifetimes and their evolutions as the degree of entanglement in these states is varied by Stark effect. These data are presented in the appended figure and are the subject of a new figure (Fig. 3) in the manuscript, together with the following text.

“The super- and sub-radiance character of the $|S\rangle$ and $|A\rangle$ states is further evidenced by measurements of their lifetimes for various degrees of entanglement. Figure 3a displays the normalized PL decay curves recorded after selective, pulsed excitation of the $|S\rangle$ state (blue curve) and $|A\rangle$ state (red curve) in the situation where the molecular detuning Δ is minimized by Stark effect. These decays are well reproduced by single exponential curves with lifetimes 6.3 ns and 11.1 ns, respectively, which are markedly different from the average lifetime 7.8 ± 0.34 ns of single molecules found in this sample, as shown in the inset of Fig. 3a. One can notice that the average decay rate $(\gamma_+ + \gamma_-)/2$ coincides with the decay rate of the uncoupled single molecules, as a signature of transfer of oscillatory strength between the subradiant and superradiant states. Moreover, we find a good agreement of these lifetimes with the computed values $(\gamma_0 \mp 2ab\gamma_{12})^{-1}$ plotted as red and blue curves in Fig. 3b.”

Fig 3 Comparison of the subradiant and superradiant lifetimes.

a, Decays curves of the subradiant and superradiant states. The laser pulses have a rise-fall time of 1 ns and a repetition rate 100 kHz. The solid curves are exponential fits with lifetimes $\tau_+ = 6.3$ ns and $\tau_- = 11.1$ ns, which are consistent with the computed subradiant lifetime $\tau_- = (\gamma_0 - 2ab\gamma_{12})^{-1} = 1.43 \gamma_0^{-1}$ and the superradiant one $\tau_+ = (\gamma_0 + 2ab\gamma_{12})^{-1} = 0.77 \gamma_0^{-1}$, using $a = 0.88$, $b = 0.47$, $\gamma_{12} = 0.35 \gamma_0$. The coefficients a and b are deduced from the diagonalization of the Hamiltonian H in the absence of laser field. The detuning $\Delta = 15 \gamma_0$ is derived from the splitting between $|S\rangle$ and $|A\rangle$ and from the coupling constant $V = 9.5 \gamma_0$ that is deduced from the fits of the excitation spectra at various excitation intensities. Inset: Histogram of the lifetime of 35 uncoupled single molecules. The blue and

red bars indicate the values of τ_+ and τ_- for the coupled pair. **b**, Evolution of τ_+ (blue circles) and τ_- (red circles) with the molecular detuning Δ which is varied by differential Stark shifts of the molecular resonances from $15 \gamma_0$ (150 Volts) to $32 \gamma_0$ (0 Volt). The solid curves are the computed values of τ_+ and τ_- . The black circles are the inverse of the average subradiant and superradiant decay rates (τ_+^{-1} and τ_-^{-1}), and coincide with the average lifetime of the uncoupled single molecules (black dashed line).

2. Selective single qubit rotation is on its own an important ingredient for a quantum information system, and is not demonstrated in the present work.

We now bring to the Supplementary Information a demonstration of selective single qubit rotation as suggested by the Reviewer. Fast and highly contrasted Rabi oscillations are observed in the fluorescence of a single molecule submitted to a transient resonant excitation (appended figure). Using a laser pulse with suited duration, efficient pumping in the excited state can be achieved within times ~ 2 ns and with a probability of 0.9 corresponding to the π -pulse fidelity. This time can even be shortened and the fidelity improved using pulses with faster rise times and larger peak intensities. Damping of the Rabi oscillations (at the rate $(1/T_1 + 1/T_2)/2$ for a resonant excitation) gives access to the optical coherence lifetime T_2 once the excited state lifetime T_1 is extracted from the fluorescence decay after the excitation cutoff. Here, T_2 is found to coincide with $2T_1$, which points to zero thermal dephasing in this host-guest molecular system at liquid helium temperatures, in agreement with the CW measurements of the ZPL widths in the fluorescence excitation spectra.

Supplementary Figure 1: Single qubit rotation.

a, Sketch of the experimental setup: A CW dye laser, optically chopped with an optical modulator produces optical pulses with a duration of 100 ns and a rise time/fall time of 500 ps (10%-90%) at a repetition rate of 100 kHz. The laser frequency is locked on a wavemeter with a digital PID to the frequency of a single emitter ZPL within 10 MHz. An avalanche photodiode combined with a start-stop acquisition card synchronized with the excitation pulses records the arrival times of the red-shifted fluorescence photons stemming from the molecule. **b**, Temporal evolution of the fluorescence signal from a single DBATT molecule upon pulsed resonant excitation (Integration time 25 min, bin width 64 ps). The fluorescence background stemming from out-of-focus molecules has been subtracted and the fluorescence signal is normalized to 1/2 at long times in order to mimic the evolution of the excited state

population. The black curve is a theoretical simulation of the excited state population, which reproduces the damped Rabi oscillations following the rising edge of the excitation pulse, with a Rabi angular frequency $11.7\gamma_0$ and an optical coherence lifetime $T_2 = 2T_1$. The excited state lifetime $T_1 = (7.39 \pm 0.02)$ ns is deduced from the exponential decay following the falling edge of the pulse.

This figure and caption are incorporated in the Supplementary Information and introduced in the revised manuscript with the sentence: “For instance, Rabi oscillations in the fluorescence of a single DBATT molecule submitted to a transient resonant excitation demonstrate the possibility to prepare the excited state with a probability 0.9 after a π -pulse and show that the coherence dephasing time reaches its upper limit given by twice the excited state lifetime (Supplementary Fig. 1).”

3. On top of this, the author suggests that the system is designed with quantum information processing in mind. With this suggestion, the readers would want to see more information about the system towards quantum information processing applications, including, qubit state preparation and readout, single qubit coherence time, single qubit gate fidelity, two-qubit gate fidelity, Bell state coherence time, and the scalability of the system and control, etc.

For example, the authors do not provide information on how these states could be used for further quantum entanglement applications. In particular, for most applications, one needs to be able to initialize and prepare the molecules in given states in a controllable fashion. However, it is unclear how one can coherently drive from $|gg\rangle$ to one of $|S\rangle$ or $|A\rangle$ deterministically in the present system. Nor is it shown how one can detect $|S\rangle$ or $|A\rangle$. Furthermore, the coherence times demonstrated here are too short compared to the oscillation times rendering them unusable for most applications.

We thank the Reviewer for these remarks, which first of all help us clarifying the main goals of our manuscript. An obvious misunderstanding comes from the awkward expression “quantum information processing” used twice in the abstract, while we did not address the realization of a quantum logic operation. We have taken this opportunity to rephrase the abstract and compact it to comply with the editorial guidelines. The abstract now reads:

“The control and manipulation of quantum-entangled states is crucial for the development of quantum technologies. A promising route is to couple solid-state quantum emitters through their optical dipole-dipole interactions. Entanglement in itself is challenging, as it requires both nanometric distances between emitters and nearly degenerate electronic transitions. Implementing hyperspectral imaging to identify pairs of coupled molecules, we reach distinctive spectral signatures of maximal entanglement of the superradiant and subradiant electronic states by tuning the molecular optical resonances with Stark effect. We demonstrate far-field selective excitation of the long-lived subradiant delocalized states with a laser field tailored in amplitude and phase. Optical nanoscopy of the entangled molecules unveils novel

spatial signatures that result from quantum interferences in their excitation pathways and reveal the location of each emitter. Controlled molecular entanglement will help deciphering more complex physical or biological mechanisms governed by the coherent coupling and realizing new quantum information schemes.”

These remarks also offer us the opportunity to give more details on the properties of our molecular system. Since the focus of our paper does not concern temporal manipulation of the entanglement of the coupled molecules, which will be the subject of another study, we discuss the two-qubit properties on the basis of information collected on single qubit rapid manipulation.

- As discussed in point#2, aromatic molecules such as DBATT embedded in molecular crystals are known to have a lifetime-limited dephasing rate ($T_2 = 2T_1$) at liquid helium temperature [Basché et al. “Single Molecule Detection, Imaging and Spectroscopy” VCH 1996; Tamarat et al. “Ten years of single molecules” J Phys Chem A 104 (2000) 1; Toninelli et al. “Single organic molecules for photonic quantum technologies” Nature Materials 20 (2021) 1615].

- We have added to the revised manuscript all key information concerning single qubit manipulation. From the Rabi oscillation displayed in the new Supplementary Figure 1, single qubit state preparation can be realized with excitation pulses prepared from a single frequency laser resonant with the molecular ZPL and using fast electro-optic or acoustic modulators. Reading the emission of a single qubit can be performed with a Fabry-Perot cavity tuned to the desired ZPL [Wrigge et al. Nature physics 4 (2008) 60].

- As discussed in point#2, single-qubit gate with high fidelity can be achieved.

- We now discuss the expected two-qubit state manipulation and gate fidelity. The fluorescence intensity autocorrelation functions recorded with a laser resonant on the $|G\rangle \rightarrow |A\rangle$ and the $|G\rangle \rightarrow |S\rangle$ transitions reflect the temporal evolution of the populations of the $|A\rangle$ and $|S\rangle$ states under pulsed excitation. When the degree of entanglement in $|A\rangle$ and $|S\rangle$ is high, these functions display markedly different Rabi frequencies (see point#1) and indicate how one can prepare a maximally entangled state after applying a laser excitation at time zero. For instance, using a laser π -pulse resonant on the $|G\rangle \rightarrow |A\rangle$ or the $|G\rangle \rightarrow |S\rangle$ transition (with suited intensity and duration), one can prepare the system in the entangled state $|A\rangle$ or $|S\rangle$. Fast entanglement/disentanglement operations on the two qubits could be achieved using short electric field pulses (Stark shifts), along similar lines to what has been done with superconducting qubits [Steffen et al. Science 313 (2006) 1423]. These further steps towards the realization of quantum logic gates are beyond the scope of the present manuscript, which brings its share of novelty and will trigger a lot of interest in the scientific community.

Therefore, rather than using the two-qubit state preparation fidelity, we find more relevant to use in the manuscript the notion of degree of entanglement in the

pure states $|A\rangle$ or $|S\rangle$, as introduced in [Abourrady et al. PRA 64 (2001) 050101(R)]. In the revised version of the manuscript, we have added the following sentences.

“As defined in Ref. [Abourrady et al. PRA 64 (2001) 050101(R)] for pure bipartite two-qubit states, the degree of entanglement P_E of the two pure states $|S\rangle$ or $|A\rangle$ is obtained from their decomposition $p|\Psi_{Bell}\rangle + \sqrt{1-p^2}e^{i\phi}|\Psi_{fact}\rangle$ into two orthogonal quantum states, $|\Psi_{Bell}\rangle$ being the corresponding Bell state, $|\Psi_{fact}\rangle$ a factorizable state, p and ϕ real numbers. The degree of entanglement is defined by $P_E = p^2 = 2|ab|$ and is identical to the concurrence [Hill & Wootters, Phys. Rev. Lett. 78 (1997) 5022]. For the pair of molecules presented in Fig. 2f, P_E reaches $(95 \pm 5)\%$ at the maximal Stark voltage. Interestingly, this value sets an upper bound to the fidelity achievable when optically preparing the system in the $|S\rangle$ or $|A\rangle$ states from the ground state $|G\rangle$, using a resonant π -pulse excitation.”

- The coherence lifetime (T_2) of $|S\rangle$ and $|A\rangle$ (nearly Bell states) obtained in this work can readily be extracted from the widths of the ZPLs in the fluorescence excitation spectra (Fig 2). Alternatively, since there is no thermal dephasing at liquid helium temperatures (see previous point #2 on single qubit coherence lifetime), we deduce from our additional lifetime measurements (see point#1) that the coherence lifetimes of $|S\rangle$ and $|A\rangle$ are respectively $2\gamma_+^{-1} = 12.6$ ns and $2\gamma_-^{-1} = 22.2$ ns. These dephasing times are expected to reach $2[\gamma_0(1 \pm \alpha)]^{-1}$ for the corresponding Bell states. While lying between γ_0^{-1} and $2\gamma_0^{-1}$ for the symmetric state, T_2 is longer for the antisymmetric state and extendable when tailoring α close to 1 [Wang et al Nature Physics 15 (2019) 483] or choosing a system displaying an intrinsically high α , such as silicon-vacancy centers in diamond [Lindner et al. New J Physics 20 (2018) 115002].

Scaling up is indeed challenging since up to now fluorescent molecules are randomly distributed in their solid hosts. Chemical synthesis methods of molecular dimers with controlled separation distance are emerging [Feofanov et al. ChemistrySelect 6 (2021) 10671] and could be extended to multimers of interacting molecules. Another routes consist in using encapsulation of fluorescent molecules in nanotubes [Cambré et al. Nature Nanotech 10 (2015) 248; Allard et al. Adv. Mater. 32 (2020) 2001419; Gaufrès et al. 8 (2014) 72], or nanoprinting methods enabling the deposition of subwavelength-sized crystals hosting a countable number of photostable and oriented molecules with subwavelength positioning accuracy [Wang et al. Nature Comm. 10 (2019) 1880]. The control of multipartite molecular entanglement could be performed using pulsed inhomogeneous Stark fields and shaped laser pulses at resonance with the ZPLs for spectral selection of the states. The revised conclusion is now enriched as follows:

“Scaling up is challenging since currently fluorescent molecules are randomly distributed in their solid hosts. Chemical synthesis methods of molecular dimers with controlled separation distance are emerging [Feofanov et al. ChemistrySelect 6 (2021) 10671] and could be extended to multimers of interacting molecules. Other routes

consist in using encapsulation of fluorescent molecules in nanotubes [Cambré et al. Nature Nanotech 10 (2015) 248; Allard et al. Adv. Mater. 32 (2020) 2001419; Gauffrès et al. 8 (2014) 72], or nanoprinting methods enabling the deposition of subwavelength-sized crystals hosting a countable number of photostable and oriented molecules with subwavelength positioning accuracy [Hail et al. Nature Comm. 10 (2019) 1880].”

4. Further comments.

We thank the Reviewer for raising these comments which will help us improving the quality of the manuscript.

Figure 1: Fig 1b is difficult to understand without axis labels. The caption mentions a green arrow in Fig. 1 c/e, but none is given in the figure.

We have added axis labels to Fig. 1b and replaced “green arrows” by “orange arrows” in the revised caption.

It would help if the authors could provide some intuition as to why the symmetry of the ZPL’s are different for Figs 1d-f depending on dipole configuration.

In the H-configuration the molecules have parallel transition dipoles that are perpendicular to the intermolecular axis, while in the J-configuration the dipoles are aligned along this axis. In the classical picture of the dipole-dipole interaction, the symmetric state $|S\rangle$ is associated with in-phase dipoles. In the H-configuration, the electric field created by one dipole is parallel and in phase opposition with the second dipole, so that the energy shift $-\mathbf{d}_2 \cdot \mathbf{E}_1(\mathbf{r}_2)$ of the $|S\rangle$ state is positive. In the J-configuration, the electric field is parallel and in phase with the dipole so that the energy shift of the $|S\rangle$ state is negative.

We have added in the caption of Fig. 1: “The symmetric state $|S\rangle$ is associated with in-phase dipoles. In the H-configuration, the electric field created by one dipole is parallel and in phase opposition with the second dipole, so that the energy shift $-\mathbf{d}_2 \cdot \mathbf{E}_1(\mathbf{r}_2)$ of the $|S\rangle$ state is positive. In the J-configuration, the electric field is parallel and in phase with the dipole so that the energy shift of the $|S\rangle$ state is negative.” and have added a scheme of the dipole orientations as inserts in Fig. 1 e,f, as shown below.

Figure 2 is confusing to follow with the multitude of plots with some overlapping information.

Fig 2 c/d - how are the two resolved? Does this use Gaussian vs Doughnut beam to resolve the two peaks?

We thank the Reviewer for raising this point. In fact, there is no overlapping information between Fig. 2a,b and Fig. 2c,d, since we think that it is important to show that we can manipulate with Stark effect the degree entanglement within the $|S\rangle$ and $|A\rangle$ states in both model configurations of molecular dipoles (H- and J- configurations).

The misunderstanding about the resolution of the spectra and type of illumination probably comes from our awkward initial choice of frequency scale used in Fig. 2c,d. The spectra of Fig. 2c,d are spectral zooms of the subradiant and superradiant ZPLs recorded for a pair of coupled molecules in the H-configuration and in the situation of maximal degree of entanglement. We had centered these spectra on zero. For clarity, we have now placed for Fig. 2c and Fig. 2d a common frequency scale centered on the two-photon transition frequency, in order to explicit the frequency offsets of the zoomed-in subradiant and superradiant resonance lines. We have also clarified in the caption that Fig. 2c,d have been recorded using a Gaussian illumination.

The inset of Fig 2b and Fig 2f are qualitatively inconsistent (the “v” opens towards higher voltages in 2b, while it is the opposite for Fig 2f). In addition, where is the two-photon transition in the inset of Fig 2b?

We thank the Reviewer for mentioning this typo. In the inset of Fig. 2.b, the voltage is swept from 0 (top) to 150 V (bottom). We have added the voltage scale to the inset of Fig. 2b. These series of spectra were recorded in the low intensity regime (excitation intensity less than the saturation intensity). The two-photon transition shows up when the intensity is raised above the saturation intensity I_s , as shown in the appended figure for the same pair of coupled molecules ($I = 3 I_s$).

It is unclear whether the electric field applied is constant or a gradient across the emitters to tune the relative frequency difference.

Is there a calibration for the Stark shift from voltage to frequency units?

The static electric field felt by the two molecules is nearly identical. To be more quantitative, we have estimated the distribution of electric field above the electrodes, using Poisson's equation. Since a 700 nm layer of SiO₂ is deposited on the electrodes, the molecules are far away from regions of strong electric field gradients (see Supplementary Fig. 1). The maximal relative variation of electric field at the position of two emitters separated by 60 nm would be 5% and below 2 % for a 20 nm separation.

One cannot establish a calibration curve for the Stark effect since for a given electric field vector, each molecule will have its own Stark shift $h\delta\nu$ connected to the changes in permanent static dipole moment $\delta\boldsymbol{\mu}$ and in polarisability tensor $\delta\boldsymbol{\alpha}$ between its ground and excited electronic states: $h\delta\nu = -\delta\boldsymbol{\mu} \cdot \mathbf{E} - \mathbf{E}\delta\boldsymbol{\alpha}\mathbf{E}/2$. The quadratic contribution related to $\delta\boldsymbol{\alpha}$ (set by the molecular volume) is very similar from one molecule to the other at the voltages used in this study. The *differential* Stark shift of two DBATT molecules, which is the relevant parameter for tailoring the degree entanglement within their $|S\rangle$ and $|A\rangle$ states, is thus essentially set by their difference in $\delta\boldsymbol{\mu}$ along the applied field. Distortions of DBATT molecules in their insertion site indeed break their centrosymmetry, leading to a residual dipole moment up to few milliDebyes. The associated linear Stark coefficient is usually less than 10 MHz/(MV/m). We have added the following sentences to the caption of the corresponding Supplementary Figure:

“Under the application of a static electric field \mathbf{E} , single DBATT molecules embedded in a naphthalene crystal often exhibit both a linear and a quadratic Stark shift $h\delta\nu$ of their optical resonance, which are connected to the changes in static dipole moment $\delta\boldsymbol{\mu}$ and in polarisability tensor $\delta\boldsymbol{\alpha}$ between their ground and excited electronic states [Brunel et al. J. Phys. Chem. A 103 (1999) 2429]:

$h\delta\nu = -\delta\boldsymbol{\mu} \cdot \mathbf{E} - \mathbf{E}\delta\boldsymbol{\alpha}\mathbf{E}/2$. The quadratic contribution related to $\delta\boldsymbol{\alpha}$ (set by the

molecular volume) is very similar among molecules at the voltages used in this study. The differential Stark shift of two DBATT molecules, which is the relevant parameter for tailoring the degree entanglement within their $|S\rangle$ and $|A\rangle$ states, is thus essentially set by their difference in $\delta\mu$ along the applied field. Distortions of DBATT molecules in their insertion site indeed break their centrosymmetry, leading to a residual dipole moment up to few milliDebyes. The associated linear Stark coefficient is usually less than $10 \text{ MHz}/(\text{MV m}^{-1})$.”

The raw Stark-tuned spectral trails of the coupled molecules presented in Fig. 2f are displayed in the appended figure and have been added to the revised Supplementary Information. The molecular detuning Δ is of the order of $\gamma_0/6$ per Volt, γ_0 being the homogenous linewidth.

In Fig 2e the linewidth is plot as a function of excitation power and extrapolated to zero power. Have the authors considered time broadening and other potential broadening mechanisms?

Spectral diffusion or pure dephasing, which would be a source of spectral broadening, is reduced for single DBATT molecules in naphthalene at liquid helium temperatures. For these molecules, negligible contributions of spectral diffusion and dephasing are consistent with the fact that the average of $\gamma_-/2\pi$ and $\gamma_+/2\pi$ coincides with the homogeneous linewidth. In the appended figure, we display a spectral trail over time (with fixed voltage) for the same pair of molecules as in Fig. 3e (excitation with a Gaussian-shaped beam with power 256 nW, 5 ms per time bin). They do not exhibit spectral diffusion.

Moreover, the Rabi oscillations of a single molecule at 2K shown above in point#2 are reproduced using a coherence lifetime being twice the excited state lifetime ($T_2 = 2T_1$), i.e. with zero dephasing.

Figure 4: The scale for the inset of Fig 4a is unclear

We have added the length of the scale bar (200 nm).

What is the coherence time limited by? Is it fundamental due to the matrix environment?

Indeed, for aromatic molecules well engaged in crystallographic sites of a molecular crystal [Ten years], as in the case of DBATT molecules inserted in a naphthalene crystalline matrix, dephasing processes vanish at 2K and the coherence lifetime T_2 reaches its upper bounder $2T_1$ determined by the excited state lifetime T_1 . This host-guest system has already proved to behave as a nearly perfect two-level quantum system in test-bench nonlinear and quantum optical experiments [Lounis et al. Phys Rev Lett 78 (1997) 3673; Wrigge et al. Nature Physics 4 (2008) 60, etc...]. Moreover, throughout this work, a very good agreement is found between the experimental data (excitation spectra, time-resolved Rabi oscillations) and the theoretical simulations without resorting to a contribution of pure dephasing in the coherence lifetime.

The Rabi oscillations between $|G\rangle$ and $|A\rangle$, and $|G\rangle$ and $|S\rangle$ are demonstrated through coincidence measurement. Is there a way to directly measure the Rabi oscillation? For example, by monitoring the fluorescence as a function of time.

As previously discussed in point #2, Rabi oscillations can be measured using resonant pulsed excitation and time synchronized fluorescence detection. The figure displayed in point#2 has been added to the Supplementary Information to demonstrate fast manipulation of the Bloch vector of a single molecule (single qubit) with a pulsed laser and Rabi oscillations with a lifetime-limited coherence time.

Fast manipulation of the entanglement of two coupled molecules can be performed with resonant pulsed excitation of the $|G\rangle \rightarrow |S\rangle$ and $|G\rangle \rightarrow |A\rangle$ ZPLs. The following figure (given for the Reviewer only) shows preliminary results where Rabi oscillations can be observed under a pulsed excitation resonant on the $|G\rangle \rightarrow |S\rangle$ transition of a pair of coupled molecules. It shows the possibility of fast preparation of the system in $|S\rangle$ starting from the ground state $|G\rangle$.

These preliminary results are beyond the scope of this manuscript and pave the way to fast and deterministic preparation and manipulation of entangled states. Such entanglement manipulation together with the realization of quantum logic gates require more experimental efforts and developments and will be the subject of another paper. We believe that our manuscript already brings many novel demonstrations and, after revision and incorporation of our latest results, now deserves publication in Nature Communication.

List of changes (in addition to the editorial changes)

1- Main text.

Title: “Tailoring the degree of entanglement in the superradiant and subradiant states of two coherently coupled quantum emitters”

Abstract rephrased and compacted (149 words):

“The control and manipulation of quantum-entangled states is crucial for the development of quantum technologies. A promising route is to couple solid-state quantum emitters through their optical dipole-dipole interactions. Entanglement in itself is challenging, as it requires both nanometric distances between emitters and nearly degenerate electronic transitions. Implementing hyperspectral imaging to identify pairs of coupled molecules, we reach distinctive spectral signatures of maximal entanglement of the superradiant and subradiant electronic states by tuning the molecular optical resonances with Stark effect. We demonstrate far-field selective excitation of the long-lived subradiant delocalized states with a laser field tailored in amplitude and phase. Optical nanoscopy of the entangled molecules unveils novel spatial signatures that result from quantum interferences in their excitation pathways and reveal the location of each emitter. Controlled molecular entanglement will help deciphering more complex physical or biological mechanisms governed by the coherent coupling and developing new quantum information schemes.”

Introduction (page 2):

“Further challenges are to manipulate the degree of entanglement in delocalized states of pairs of molecules having frozen geometries and dipole orientations, and selectively address any quantum entangled state. Coherent and dissipative dipole-dipole interactions give rise to collective phenomena of super- or subradiance¹⁷, in which a collective excitation of the emitters decays faster or slower than the individual molecular excitations, respectively. While superradiance has been widely studied since the pioneering work of Dicke¹⁸, few experimental studies have been reported on subradiance, using essentially ultracold atoms and molecules¹⁹⁻²², metamaterial lattices²³ or single structured atomic layers acting as optical mirrors²⁴.”

Introduction (page 3):

“...we demonstrate the manipulation of the degree of entanglement in their delocalized electronic states through Stark shifts of their optical resonances. Direct evidence of subradiance (superradiance) is brought through lengthening (shortening) of the fluorescence lifetime recorded when the laser is tuned to the corresponding state. Nearly pure Bell states³³ are achieved and delocalized molecular electronic states are found...”

Page 3:

“For instance, Rabi oscillations in the fluorescence of a single DBATT molecule submitted to a transient resonant excitation demonstrate the possibility to prepare the excited state with a probability 0.9 after a π -pulse and show that the coherence dephasing time reaches its upper limit given by twice the excited state lifetime (Supplementary Fig. 1).”

Page 4:

“The vacuum-induced coherent coupling between two molecules...”

Figure 1:

We have added a scheme of the dipole orientations as inserts in Fig. 1 e,f, as shown below.

In its caption, we have added: “The symmetric state $|S\rangle$ is associated with in-phase dipoles. In the H-configuration, the electric field created by one dipole is parallel and in phase opposition with the second dipole, so that the energy shift $-\mathbf{d}_2 \cdot \mathbf{E}_1(\mathbf{r}_2)$ of the $|S\rangle$ state is positive. In the J-configuration, the electric field is parallel and in phase with the dipole so that the energy shift of the $|S\rangle$ state is negative.” We have also done the correction: “...orange arrows”...

Page 5:

“...the degree of entanglement in the states $|S\rangle$ and $|A\rangle$ can be tuned only by adjusting their resonance frequencies. Manipulation of this degree of entanglement by Stark shifting...”

Page 6:

“...the reduced energy splitting $2V$ between the two entangled states obtained at the minimal detuning $|\Delta|$ is a signature of a weak dipole-dipole coupling ($V \sim \gamma_0$), which means that the coherent dipole-dipole interaction can create delocalized entangled states between solid-state quantum emitters separated by a distance as large as ~ 60 nm, using $V = -3\alpha\gamma_0/2(kr_{12})^3$.”

“Maximal entanglement in the states $|S\rangle$ and $|A\rangle$ is also demonstrated... Voltage manipulation of the degree of entanglement in $|S\rangle$ and $|A\rangle$ is presented...”

Page 7:

“As defined in Ref.⁴⁶ for pure bipartite two-qubit states, the degree of entanglement P_E of the two pure states $|S\rangle$ or $|A\rangle$ is obtained from their decomposition $p|\Psi_{Bell}\rangle + \sqrt{1-p^2}e^{i\phi}|\Psi_{fact}\rangle$ into two orthogonal quantum states, $|\Psi_{Bell}\rangle$ being the corresponding Bell state, $|\Psi_{fact}\rangle$ a factorizable state, p and ϕ real numbers. The degree of entanglement is defined by $P_E = p^2 = 2|ab|$ and is identical to the concurrence⁴⁷. For the pair of molecules presented in Fig. 2f, P_E reaches $(95 \pm 5)\%$ at the maximal Stark voltage. Interestingly, this value sets an upper bound to the fidelity achievable when optically preparing the system in the $|S\rangle$ or $|A\rangle$ states from the ground state $|G\rangle$, using a resonant π -pulse excitation.”

Modifications in Figure 2b,c,d,e:

The caption title becomes: “Manipulation of the degree of entanglement by Stark effect.” The modifications in the caption are: “The inset in **b** displays the spectral trails of this pair as the voltage is swept from 0 (top) to 150 V (bottom). **c,d**, Zoomed-in ZPLs...” and “The experimental spectra have been recentered on the two-photon transition, while the raw spectral trails are presented in Supplementary Fig. 7.”

New figure (Figure 2) added, together with a paragraph on pages 8-9.

“The super- and sub-radiance character of the $|S\rangle$ and $|A\rangle$ states is further evidenced by measurements of their lifetimes for various degrees of entanglement. Figure 3a displays the normalized PL decay curves recorded after selective, pulsed excitation of

the $|S\rangle$ state (blue curve) and $|A\rangle$ state (red curve) in the situation where the molecular detuning Δ is minimized by Stark effect. These decays are well reproduced by single exponential curves with lifetimes 6.3 ns and 11.1 ns, respectively, which are markedly different from the average lifetime 7.8 ± 0.34 ns of single molecules found in this sample, as shown in the inset of Fig. 3a. One can notice that the average decay rate $(\gamma_+ + \gamma_-)/2$ coincides with the decay rate of the uncoupled single molecules, as a signature of transfer of oscillatory strength between the subradiant and superradiant states. Moreover, we find a good agreement of these lifetimes with the computed values $(\gamma_0 \mp 2ab\gamma_{12})^{-1}$ plotted as red and blue curves in Fig. 3b.”

Fig 3 Comparison of the subradiant and superradiant lifetimes.

a, Decays curves of the subradiant and superradiant states. The laser pulses have a rise-fall time of 1 ns and a repetition rate 100 kHz. The solid curves are exponential fits with lifetimes $\tau_+ = 6.3$ ns and $\tau_- = 11.1$ ns, which are consistent with the computed subradiant lifetime $\tau_- = (\gamma_0 - 2ab\gamma_{12})^{-1} = 1.43 \gamma_0^{-1}$ and the superradiant one $\tau_+ = (\gamma_0 + 2ab\gamma_{12})^{-1} = 0.77 \gamma_0^{-1}$, using $a = 0.88$, $b = 0.47$, $\gamma_{12} = 0.35 \gamma_0$. The coefficients a and b are deduced from the diagonalization of the Hamiltonian H in the absence of laser field. The detuning $\Delta = 15 \gamma_0$ is derived from the splitting between $|S\rangle$ and $|A\rangle$ and from the coupling constant $V = 9.5 \gamma_0$ that is deduced from the fits of the excitation spectra at various excitation intensities. Inset: Histogram of the lifetime of 35 uncoupled single molecules. The blue and red bars indicate the values of τ_+ and τ_- for the coupled pair. **b**, Evolution of τ_+ (blue circles) and τ_- (red circles) with the molecular detuning Δ which is varied by differential Stark shifts of the molecular resonances from $15 \gamma_0$ (150 Volts) to $32 \gamma_0$ (0 Volt). The solid curves are the computed values of τ_+ and τ_- . The black circles are the inverse of the average subradiant and superradiant decay rates (τ_+^{-1} and τ_-^{-1}), and coincide with the average lifetime of the uncoupled single molecules (black dashed line).

New Figures 4c,d added, together with sentences and caption on page 9-11:

“Striking differences in coupling of $|S\rangle$ and $|A\rangle$ to the laser field manifest in the fluorescence intensity autocorrelation function $g^{(2)}(\tau)$ recorded under resonant excitation of $|S\rangle$ and $|A\rangle$ at identical laser intensities. Indeed, besides strong photon antibunching on both transitions, the normalized coincidence histograms show Rabi oscillations that are much faster on the $|G\rangle \rightarrow |S\rangle$ transition (Fig. 4a) than on the $|G\rangle \rightarrow |A\rangle$ one (Fig. 4b) when the degree of entanglement in $|S\rangle$ and $|A\rangle$ is maximal. In contrast, the Rabi frequencies become similar when lowering the degree of entanglement by Stark shifting these molecular resonances apart (Fig. 4c,d). (...)

Incidentally, photon bunching¹⁶ is evidenced when the laser is tuned to the central two-photon resonance (Supplementary Fig. 10), as a signature of simultaneous excitation of coherently coupled molecules to the doubly excited state $|E\rangle$ with the subsequent two-photon emission cascade.”

a,b, Normalized photon coincidence histograms (after background subtraction) measured for a pair of coupled molecules excited with a Gaussian-shaped laser beam of intensity 200 W cm^{-2} at resonance with the $|G\rangle \rightarrow |S\rangle$ transition (**a**) and with the $|G\rangle \rightarrow |A\rangle$ transition (**b**) when the degree of entanglement is maximized by Stark effect ($P_E \sim 0.92$, voltage 150 V). They are characterized by an antibunching dip and damped Rabi oscillations, while photon bunching is evidenced when the laser is resonant on the two-photon transition (Supplementary Fig. 10). The second-order correlation functions (red curves) are computed for nearly parallel dipoles in the J-configuration with $V = -17 \gamma_0$, $\gamma_{12} = 0.3 \gamma_0$, and $\Delta = 10 \gamma_0$ (Supplementary Note 1). **c,d**, Photon coincidence histograms obtained for the same pair of coupled molecules when the molecular resonances are tuned apart by Stark effect (-150 V), leading to a modest degree of entanglement $P_E \sim 0.24$. The excitation intensity is 40 W cm^{-2} . The simulations are performed with the same parameters except for $\Delta = 60 \gamma_0$.

Modifications in Figure 5 (previously Figure 4):

We have added the length of the scale bar (200 nm).

Conclusion (Page 13-14): The discussion has been extended as follows.

“In summary, we demonstrate the manipulation of the degree of entanglement in the superradiant and subradiant states of coherently coupled solid-state quantum emitters, using Stark shifts of their electronic levels. Nearly maximal quantum entanglement is achieved for emitters separated by distances up to several tens of nanometers in various dipole configurations. Moreover, a simple far-field optical method is developed to selectively excite the super- and subradiant states, which opens new opportunities in quantum information schemes, in particular those exploiting the long-lived radiative

decay of subradiant states. Novel spatial signatures inherent to dipole-dipole interaction are unveiled with ESSat nanoscopy and provide a unique mean to localize the coupled emitters. This far-field versatile method could be extended to engineer the spectroscopic selection rules for various molecular or nanometer scale systems, or to perform fast manipulation of entanglement for quantum logic gate operations⁵³. The present study also opens up the opportunity to investigate the rich quantum signatures of the light emitted from collective delocalized excitations, such as the $|E\rangle \rightarrow |S\rangle \rightarrow |G\rangle$ cascade that can be used to generate time-energy entangled photon-pairs^{54,55}.

Future investigation of the entanglement in systems scaled up to arrays of two-level emitters should focus on highly directional scattering properties²⁴, highly nonlinear responses with few photons⁵⁶ and should set the foundations for novel platforms of light-matter interfaces. Scaling up is challenging since currently fluorescent molecules are randomly distributed in their solid hosts. Chemical synthesis methods of molecular dimers with controlled separation distance are emerging⁵⁷ and could be extended to multimers of interacting molecules. Other routes consist in using encapsulation of fluorescent molecules in nanotubes⁵⁸⁻⁶⁰, or nanoprinting methods enabling the deposition of subwavelength-sized crystals hosting a countable number of photostable and oriented molecules with subwavelength positioning accuracy⁶¹.

The experimental and theoretical tools developed here to study the test-bench entangled molecular pair should also foster thorough investigations of a wealth of elaborate coupled systems, such as polymer conjugates⁶² and quantum dot molecules⁹. It will be particularly interesting to explore the decoherence processes in highly delocalized molecular systems. For instance, light-harvesting complexes⁶³ are composed of a very dense set of fluorophores separated by few nanometers, leading to extremely strong dipolar coupling between molecules and multipartite quantum entanglement that survives over picosecond timescales at room temperature despite decoherence effects associated with the surrounding phonon bath⁶⁴⁻⁶⁶. The robustness of these quantum systems against decoherence mechanisms are still poorly understood. Open questions concern the coupling of such delocalized states to phonons and the possible existence of non-Markovian effects. A pair of coupled molecules can therefore be the ideal elementary brick to help deciphering the decoherence mechanisms in these complex systems.”

Data availability:

All relevant data that support our experimental findings are available from the corresponding author upon reasonable request.

Acknowledgement:

We have added this acknowledgement paragraph. “We acknowledge the financial support from the French National Agency for Research, Région Aquitaine, Institut Universitaire de France, Idex Bordeaux (LAPHIA Program), and the EUR Light S&T Graduate Program (PIA3 Program “Investment for the Future”, ANR-17-EURE- 0027), and GPR LIGHT. We are grateful to E. Cormier for loaning a fast pulse generator, and D. Maily (C2N) for help in the fabrication of the electrodes.”

Added/removed references:

Removed:

Gao et al. Nature Photonics 9, 363–373 (2015).

Agarwal, Tracts in Modern Physics, Springer Verlag, 1974

Added:

Hepp et al. Advanced Quantum Technologies 1900020 (2019).

Toninelli, C. et al. Nat Mater 20, 1615–1628 (2021).

van Loo et al. Science 342, 1494–1496 (2013).

Dicke, R. H. Phys. Rev. 93, 99 (1954).

Guerin et al. Physical Review Letters 116, 083601 (2016).

DeVoe et al. Physical Review Letters 76, 2049–2052 (1996).

Takasu et al. Physical Review Letters 108, 173002 (2012).

McGuyer et al. Nature Physics 11, 32–36 (2014).

Jenkins et al. Physical Review Letters 119, 053901 (2017).

Rui et al. Nature 583, 369–374 (2020).

Abouraddy et al. Physical Review A 64, 050101(R) (2001).

Hill & Wootters, Physical Review Letters 78, 5022–5025 (1997).

Simon & Poizat, Physical Review Letters 94, 030502 (2005).

Jayakumar et al., Nature Communications 5, 4251 (2019).

Holzinger et al., Appl. Phys. Lett. 1–8 (2021)

Feofanov et al., Nature Physics 6, 593–597 (2010).

Cambré et al., Nature Nanotechnology 10, 248–252 (2015).

Allard et al., Advanced Materials 32, 2001429 (2020).

Gaufrès et al., Nature Photonics 8, 72–78 (2014).

Hail et al., Nature Communications 10, 1880 (2019).

Sarovar et al., Nature Physics 6, 462–467 (2010).

Collini et al., Nature 463, 644–647 (2010).

Chin et al., Nature Physics 9, 113–118 (2013)

2- Supporting information:

New figure: Supplementary Figure 1.

Supplementary Figure 1: Single qubit rotation.

a, Sketch of the experimental setup: A CW dye laser, optically chopped with an optical modulator produces optical pulses with a duration of 100 ns and a rise time/fall time of 500 ps (10%-90%) at a repetition rate of 100 kHz. The laser frequency is locked on a wavemeter with a digital PID to the frequency of a single emitter ZPL within 10 MHz. An avalanche photodiode combined with a start-stop acquisition card synchronized with the excitation pulses records the arrival times of the red-shifted fluorescence photons stemming from the molecule. **b**, Temporal evolution of the fluorescence signal from a single DBATT molecule upon pulsed resonant excitation (Integration time 25 min, bin width 64 ps). The fluorescence background stemming from out-of-focus molecules has been subtracted and the fluorescence signal is normalized to 1/2 at long times in order to mimic the evolution of the excited state population. The black curve is a theoretical simulation of the excited state population, which reproduces the damped Rabi oscillations following the leading edge of the excitation pulse, with a Rabi angular frequency $11.7 \gamma_0$ and an optical coherence lifetime $T_2 = 2T_1$. The excited state lifetime $T_1 = (7.39 \pm 0.02)$ ns is deduced from the exponential decay following the trailing edge of the pulse.

Modifications in Supplementary Figure 2 (previously named Supplementary Figure 1).

We have indicated the thickness of the SiO₂ layer (0.7 μm) on the Figure and in the caption. We also have added information about the Stark shifts:

“Under the application of a static electric field \mathbf{E} , single DBATT molecules embedded in a naphthalene crystal often exhibit both a linear and a quadratic Stark shift $h\delta\nu$ of their optical resonance, which are connected to the changes in static dipole moment $\delta\mu$ and in polarisability tensor $\delta\alpha$ between their ground and excited electronic states [Brunel et al. J. Phys. Chem. A 103 (1999) 2429]:

$h\delta\nu = -\delta\boldsymbol{\mu}\cdot\mathbf{E} - \mathbf{E}\delta\alpha\mathbf{E}/2$. The quadratic contribution related to $\delta\alpha$ (set by the molecular volume) is very similar among molecules at the voltages used in this study. The differential Stark shift of two DBATT molecules, which is the relevant parameter for tailoring the degree entanglement within their $|S\rangle$ and $|A\rangle$ states, is thus essentially set by their difference in $\delta\boldsymbol{\mu}$ along the applied field. Distortions of DBATT molecules in their insertion site indeed break their centrosymmetry, leading to a residual dipole moment up to few milliDebyes. The associated linear Stark coefficient is usually less than $10\text{ MHz}/(\text{MV m}^{-1})$.”

New figure: Supplementary Figure 7.

Supplementary Figure 7: Raw Stark-tuned spectral trails.

These spectral trails are built from the raw fluorescence excitation spectra measured on the coupled molecules presented in Fig. 2f. They display quadratic and linear Stark shifts of the molecular resonances, leading to a net detuning of the molecular resonances and signatures of a variation of the degree of entanglement in the superradiant and subradiant states.

New figure: Supplementary Figure 7.

Supplementary Figure 10: Photon coincidence histograms under selective excitation of the superradiant, subradiant and doubly excited states.

a,b,c, Normalized photon coincidence histograms measured on the pair of coupled molecules presented in Fig. 4 when the Gaussian-shaped laser beam of intensity 200 W cm^{-2} is tuned at resonance with the symmetric state (**a**), the two-photon transition (**b**) and the antisymmetric state (**c**). Photon bunching is evidenced in (**c**) as a signature of two-photon emission from the doubly excited state $|E\rangle$. The second-order correlation functions (red curves) are computed for nearly parallel dipoles in the J-configuration, $V = -17 \gamma_0$, $\gamma_{12} = 0.3 \gamma_0$, and $\Delta = 10 \gamma_0$. The black arrows indicate the correspondence between the coincidence histograms and the selected ZPLs in the fluorescence excitation spectrum (**d**) recorded at the same intensity.

Additional Reference:

Brunel et al., J. Phys. Chem. A 103, 2429–2434 (1999).

REVIEWER COMMENTS

Reviewer #2 (Remarks to the Author):

The authors treat the review comments very seriously and carefully. All the comments are well addressed so that the manuscript is improved a lot. I suggest an acceptance of the paper in the present form.

Reviewer #3 (Remarks to the Author):

The revised manuscript is much improved, but some concerns that I had were not addressed sufficiently. First, I found the language of “degree of entanglement” cumbersome and misleading. As I had written previously, a direct measurement of the off-diagonal terms in the two-qubit density matrix is required to claim entanglement. A measurement such as state tomography or parity oscillations as shown in [Phys. Rev. A 82, 030306(R), Phys. Rev. Lett. 81, 3631–3634 (1998)] is needed. I understand that this is not what the current study is capable of, and I therefore suggest removing the claim or the usage of entanglement as much as possible. Using terms such as superradiant and subradiant states are sufficient and clear. I suggest the title can simply be “tailoring superradiant and subradiant nature of two coherent...” Furthermore, no information is given regarding selective manipulation and readout of a single molecule within the entangled molecule pair, which is necessary for any platform hoping to take advantage of the quantum properties. While the authors in the rebuttal have included a plot for single molecule Rabi flopping, it is unclear whether this measurement is performed on a single molecule within a closely spaced pair of molecules, or if this is on an isolated molecule completely. This should be clarified.

From my perspective, it seems challenging to turn off the interactions between two molecules at will to change between two molecules state manipulation and single molecule state manipulation in this platform. Would it be possible to use the E-field to tune the detuning to shift between coupled and uncoupled states? Would such an adiabatic ramping process be too slow to maintain any coherence between the two molecules? I suggest the authors address this point in the manuscript which is relevant to claim such a system could be a quantum resource.

The authors have included in the conclusion potential applications towards studies of photochemical processes, but it is not clear what sorts of requirements a platform would need to satisfy to be able to study those systems. Perhaps the author can educate the readers.

In addition, a few more points came up in the revised manuscript that require further clarification.

1. For the fidelity, the equation that is used seems to assume a pure state to start with. Is this a valid assumption? The authors should elaborate on the assumptions made and their validity.

2. In the paper the authors demonstrate selective preparation of the superradiant and subradiant states using Gaussian vs donut shaped beams. While I can see how this could be useful to prepare the two states where they are not clearly resolved, such as in Fig. 2b of the main manuscript, the data shown in Fig. 4e-f is chosen for a location where the detunings are fairly large and are already well-resolved. Is there any reason the authors chose such a location? In addition, why is there a discrepancy between the theory simulation with Gaussian illumination in Fig 2 with that shown in Fig 4e-f under similar conditions (splitting at roughly $20 \cdot \omega / \gamma_0$). Finally, it would be informative if the authors could elaborate how the donut beam can address emitters spaced a distance much less than the diffraction limit apart.

3. In the manuscript the authors explain the origin of the differential Stark shift between the molecules. The number that is given is $\sim 10 \text{ MHz}/(\text{MV}/\text{cm})$, with the couple of MV/cm quoted in the manuscript, I am wondering how the differential shifts add up to 100's of MHz.

4. While the linewidth of the subradiant and superradiant states are in the range of 10s MHz, the observed T1 times are only a couple of ns. Naively we would expect $\sim 100 \text{ MHz}$ linewidth if the lifetime of the state is $\sim 10 \text{ ns}$. Could the authors comment on how the two numbers work out?

Reviewer #4 (Remarks to the Author):

I have been asked an opinion concerning the manuscript by J.B. Trebbia et al., which has already undergone three reviewers reports. In my opinion, the manuscript reports significant results, which are worth being published in Nature Communications.

The authors have experimentally shown that it is possible to control the coherent coupling of two dipole-coupled molecules, by application of an external electric field that allows to tune the symmetric and antisymmetric states energy splitting by Stark effect. The two states lead to superradiant (reduced lifetime) or subradiant (increased lifetime) emission. A novel far-field coupling technique has been developed to selectively excite one or the other. In my opinion, these results set a relevant step towards

application of molecular states for quantum information processing, although scalability is a real challenge and it is not totally clear how it is going to be faced.

The manuscript is very well written and thoroughly referenced, several details are also given in the supplementary file. The experimental results are robust, and their interpretation correct, as far as I can judge. After going over the previous comments and concerns from the reviewers, and the authors' rebuttal, I think the latter have answered most appropriately to all the criticism raised. In particular, the most significant points raised from Reviewer 1 have been satisfactorily addressed, in my opinion, and the amendments to the manuscript make this revised version suited for publication. In particular, I tend to agree with the authors that the realization of quantum logic gates would require significantly more experimental effort that goes beyond the present scope of their manuscript. Such a result would probably deserve an even more impactful publication.

I only have minor, mostly stylistic comments the authors might want to keep into account.

- the onset of two-photon transition in panel 1b should be explicitly indicated, for easier reading;

- it is not clear why in Fig. 3b the black and blue curves have only 2 points for splittings close to 0.5 and 0.8 GHz, while the red curve has several intermediate points as well.

Color code:

Blue for the Reviewers' remarks;

Black for the Authors' response;

Green for the modifications in the manuscript or Supplementary Information.

Response to Reviewer# 3

We thank the Reviewer for noticing the large improvement of our manuscript. We address below all the remaining concerns of the Reviewer.

The revised manuscript is much improved, but some concerns that I had were not addressed sufficiently. First, I found the language of “degree of entanglement” cumbersome and misleading. As I had written previously, a direct measurement of the off-diagonal terms in the two-qubit density matrix is required to claim entanglement. A measurement such as state tomography or parity oscillations as shown in [Phys. Rev. A 82, 030306(R), Phys. Rev. Lett. 81, 3631–3634 (1998)] is needed. I understand that this is not what the current study is capable of, and I therefore suggest removing the claim or the usage of entanglement as much as possible. Using terms such as superradiant and subradiant states are sufficient and clear. I suggest the title can simply be “tailoring superradiant and subradiant nature of two coherent...”

We followed the suggestion and replaced “entanglement” by “**delocalization**” as much as possible in the text, leaving entanglement for the general introduction and outlook. The title is changed and now reads “Tailoring the degree of **delocalization** in the superradiant and subradiant states of two coherently coupled quantum emitters”.

Furthermore, no information is given regarding selective manipulation and readout of a single molecule within the entangled molecule pair, which is necessary for any platform hoping to take advantage of the quantum properties. While the authors in the rebuttal have included a plot for single molecule Rabi flopping, it is unclear whether this measurement is performed on a single molecule within a closely spaced pair of molecules, or if this is on an isolated molecule completely. This should be clarified.

In the previous rebuttal letter as well as in the Supplementary Information, we have indeed added a figure on the temporal evolution of the fluorescence signal from a single DBATT molecule upon pulsed resonant excitation. We have now specified that this molecule is **isolated** in the main text and in the figure caption. Selective manipulation and readout of a single molecule within an entangled molecule pair will require prior disentanglement, for instance using large Stark shifts so that that $|\Delta| \gg |V|$, which is readily achievable given the Stark shifts (see the point#3). Such experiments are beyond the scope of this manuscript.

From my perspective, it seems challenging to turn off the interactions between two molecules at will to change between two molecules state manipulation and single molecule state manipulation in this platform. Would it be possible to use the E-field to tune the detuning to shift between coupled and uncoupled states? Would such an adiabatic ramping process be too slow to maintain any coherence between the two molecules? I suggest the authors address this point in the manuscript which is relevant to claim such a system could be a quantum resource.

As stated in the manuscript, the coupling constant is *fixed* for two molecules rigidly embedded in a solid matrix (regardless of the applied electric field). A key point of our experiment is the ability to tune the degree of electronic-state delocalization by Stark shifting

the molecular transitions. Adiabatic passage between electronic states with Stark effect requires that the sweeping rate of the energy levels be lower than the coupling constant. An advantage of molecular systems lies in the fact that the separation distance between molecules can be as short as few nanometers. Therefore, the coupling constant of molecular systems can exceed 1 GHz, enabling adiabatic sweeping times in the nanoseconds range. These sweeping times will be much shorter than the coherence lifetimes (~20 ns), allowing rapid, adiabatic manipulation of the entanglement for quantum logic gates operations. Moreover, schemes of two-qubit logic gates operated at rates higher than the adiabatic regime [Ref. 53 of the manuscript and references therein] can be envisaged.

In the conclusion, we have added the following sentence:

“The present study opens up the opportunity to perform fast manipulation of entanglement for quantum logic gate operations⁵³, using rapid entanglement-disentanglement operations on molecular qubits with short electric field pulses.”

The authors have included in the conclusion potential applications towards studies of photochemical processes, but it is not clear what sorts of requirements a platform would need to satisfy to be able to study those systems. Perhaps the author can educate the readers.

A pair of coupled molecules can be the ideal elementary brick to help deciphering the decoherence mechanisms in more complex delocalized molecular systems, for instance by studying the temperature dependence of the degree of delocalization. Indeed, the robustness of these quantum systems against decoherence mechanisms are still poorly understood, with open questions concerning the coupling of such delocalized states to phonons and the possible existence of non-Markovian effects. Recently, a theoretical work has predicted that interaction of two strongly coupled qubits with dephasing reservoirs can lead to a long-lived mixed entangled state [Vovcenko et al. *Optics Express* 29 (2021) 9685].

In the conclusion, we have rephrased the last sentences as follows and added two references:

“Open questions concern the coupling of such delocalized states to phonons [Monshouwer et al. *J. Phys. Chem. B* 101 (1997) 7241; Vovcenko et al. *Optics Express* 29 (2021) 9685] and the possible existence of non-Markovian effects. With the aim of deciphering the decoherence mechanisms in these complex systems, a pair of coupled molecules can therefore be the ideal test-bench elementary brick, starting with the temperature dependence of its degree of entanglement”.

In addition, a few more points came up in the revised manuscript that require further clarification.

1. For the fidelity, the equation that is used seems to assume a pure state to start with. Is this a valid assumption? The authors should elaborate on the assumptions made and their validity.

We would like first to recall that we do not claim any temporal manipulation of the entanglement in the manuscript and SI, so we do not understand which start-state the reviewer is referring to. In the paragraph discussing the degree of entanglement (now called degree of delocalization, page7) P_E of the two *pure* states $|S\rangle$ or $|A\rangle$ (that are eigenstates of the Hamiltonian describing the coupled molecular system), we only refer to the theoretical upper

value of the fidelity that one can expect in an experiment aiming at preparing a Bell state with a π -optical pulse, starting from the pure ground state $|G\rangle$.

Maybe the reviewer refers to our previous rebuttal letter where we have discussed the fidelity of a *single qubit* preparation with a π -optical pulse, starting from the pure ground state of the single qubit. We have studied an isolated molecule initially in its pure ground state $|g\rangle$, and measured the temporal evolution of the fluorescence signal (proportional to the population of the excited state $|e\rangle$) when the molecule is submitted to a resonant laser intensity step. The fidelity is given by the amplitude of the first population maximum of the temporal evolution.

2. In the paper the authors demonstrate selective preparation of the superradiant and subradiant states using Gaussian vs donut shaped beams. While I can see how this could be useful to prepare the two states where they are not clearly resolved, such as in Fig. 2b of the main manuscript, the data shown in Fig. 4e-f is chosen for a location where the detunings are fairly large and are already well-resolved. Is there any reason the authors chose such a location? In addition, why is there a discrepancy between the theory simulation with Gaussian illumination in Fig 2 with that shown in Fig 4e-f under similar conditions (splitting at roughly $20 \cdot \omega / \gamma_0$). Finally, it would be informative if the authors could elaborate how the donut beam can address emitters spaced a distance much less than the diffraction limit apart.

The interaction of the coupled molecular system with the driving laser field is given by the third term of Eq. 1. For a maximally entangled state (Bell state, corresponding to $a = b = 1/\sqrt{2}$) one directly sees that excitation of the subradiant state with a conventional Gaussian-shaped laser field having identical amplitudes and phases at the molecular positions ($\Omega_1 = \Omega_2$) is forbidden, regardless of the spectral separation between $|S\rangle$ and $|A\rangle$. This is why we have implemented a simple, far-field method of efficient and selective excitation of $|A\rangle$. Using a donut-shaped beam, whose field extends over a distance much larger than the molecular separation distance (as noticed by the Reviewer), we demonstrate selective excitation of the $|A\rangle$ state, provided that its zero-field center is exactly placed midway between both emitters. Indeed, in this situation the doughnut laser field has identical amplitudes and opposite phases at the molecules ($\Omega_1 = -\Omega_2$), as illustrated in the inset of Fig. 4e.

To make this key point stronger in the manuscript, we have rephrased the corresponding paragraph:

“Considering a parallel dipole configuration, excitation of the subradiant state with a conventional Gaussian-shaped laser field having identical amplitudes and phases at the molecular positions ($\Omega_1 = \Omega_2$) is forbidden according to Eq. 1. However, shaping the field with identical amplitudes and opposite phases at the molecular positions ($\Omega_1 = -\Omega_2$) will enable the excitation of the subradiant state and forbid that of the symmetric superradiant state. We show here that the excitation of the sole antisymmetric subradiant state can be achieved, using a circularly polarized doughnut-shaped (first order Laguerre-Gaussian) beam whose zero-field center is exactly placed midway between both emitters”.

There is no discrepancy between theory and experiment. The simulations of Fig. 4e reproduce well the experimental spectra of Fig. 4f. The pairs of molecules presented in Fig. 2 and Fig. 4e-f are different.

3. In the manuscript the authors explain the origin of the differential Stark shift between

the molecules. The number that is given is ~ 10 MHz/(MV/cm), with the couple of MV/cm quoted in the manuscript, I am wondering how the differential shifts add up to 100's of MHz.

The Reviewer probably made confusion between centimeter and meter. Indeed, the number that is given is 10 MHz/(MV m⁻¹) in the caption of Supplementary Figure 2. This is consistent with the Stark-tuned experimental spectra. Indeed, since the electrodes are separated by 10 μ m, a variation of applied voltage of 300 Volts corresponds to an electrostatic field variation of 30 MV/m, which may lead to single-molecule Stark shifts of few hundreds of MHz. Moreover, the molecules of a coupled pair may exhibit linear Stark shift components with opposite signs, leading to differential shifts of hundreds of MHz.

4. While the linewidth of the subradiant and superradiant states are in the range of 10 s MHz, the observed T_1 times are only a couple of ns. Naively we would expect ~ 100 MHz linewidth if the lifetime of the state is ~ 10 ns. Could the authors comment on how the two numbers work out?

Spectral linewidths in the frequency domain are not obtained by taking the inverse of the lifetime: A factor of 2π is involved.

Indeed, the homogeneous absorption line-shape of an electric dipole oscillator is given by the Fourier transform of the autocorrelation function of the transition dipole moment [R. Kubo in "Fluctuation, Relaxation and Resonance in Magnetic Systems" Ed.: D. Ter Harr, Oliver and Boyd, Edimburgh, 1965]. Since the autocorrelation function has an exponential decay with time constant T_2 , its Fourier transform is a Lorentzian profile with FWHM $2/T_2$ in the *angular frequency domain*. Thus, the homogeneous width is $\Delta\nu = \frac{1}{2\pi} \times \frac{2}{T_2}$ in the *frequency domain*. The coherence lifetime of the transition dipole T_2 is related to the excited state lifetime T_1 by $1/T_2 = 1/2T_1 + 1/T_2^*$, where $1/T_2^*$ is the pure dephasing rate. Since $1/T_2^* = 0$ for single aromatic molecules in crystalline matrices at liquid helium temperatures, their ZPL-FWHM expressed in frequency units reaches their lower bound $1/2\pi T_1$ [Basché et al. "Single Molecule Optical Detection, Imaging and Spectroscopy", VCH 1996]. For DBATT molecules in naphthalene at liquid helium temperatures, the lifetime $T_1 \sim 8$ ns leads to ZPL widths of ~ 20 MHz [Jelesko et al. J. Chem. Phys. 107 (1997) 1692].

Response to Reviewer# 4

We thank the Reviewer for enthusiastic comments about the quality and novelty of our work, and recommendation of this paper for publication in Nature Communication. We address below the minor comments raised by the Reviewer.

- the onset of two-photon transition in panel 1b should be explicitly indicated, for easier reading;

We have now labeled the transitions $|G\rangle \rightarrow |S\rangle$, $|G\rangle \rightarrow |E\rangle$ and $|G\rangle \rightarrow |A\rangle$ on Fig. 1b. We also indicate in the figure caption that the two-photon transition refers to $|G\rangle \rightarrow |E\rangle$.

- it is not clear why in Fig. 3b the black and blue curves have only 2 points for splittings close to 0.5 and 0.8 GHz, while the red curve has several intermediate points as well.

These data points have been collected on the same pair of coupled molecules, starting with resonant excitation of the subradiant state, then on the superradiant state. Unfortunately, in the course of these time-consuming measurements, the couple of molecules have been lost, due to a spectral jump or a loss of optical alignment.

REVIEWERS' COMMENTS

Reviewer #3 (Remarks to the Author):

The demonstration of superradiant and subradiant states is convincing and nice, but there is no need to oversell the work. Changing the wording from “entanglement” to “delocalization” in the present version of the manuscript is confusing and inaccurate. I suggest the wording for the title “Creating superradiant and subradiant states of two coherently coupled quantum emitters”, or my original proposal of “Tailoring the superradiant and subradiant nature of two coherently coupled quantum emitters”. In addition I suggest, wherever the “degree of entanglement” or similar concept is mentioned in the manuscript, to use the wording “superposition” instead. Without a means to separate then measure the individual molecules from an entangled state, the claim of entanglement is misleading. Therefore, I suggest distinguishing between the use of “entangled state”, which is ok in this case, and referring to the “entanglement” as a defined resource, which is not ok in this case. After these changes, I recommend the paper for publication in Nature Communications.

Color code:

Blue for the Reviewers' remarks;

Black for the Authors' response;

Green for the modifications in the manuscript or Supplementary Information.

Response to Reviewer# 3

We thank the Reviewer for careful reading of our manuscript. We address below the remaining concerns of the Reviewer.

The demonstration of superradiant and subradiant states is convincing and nice, but there is no need to oversell the work. Changing the wording from “entanglement” to “delocalization” in the present version of the manuscript is confusing and inaccurate. I suggest the wording for the title “Creating superradiant and subradiant states of two coherently coupled quantum emitters”, or my original proposal of “Tailoring the superradiant and subradiant nature of two coherently coupled quantum emitters”. In addition I suggest, wherever the “degree of entanglement” or similar concept is mentioned in the manuscript, to use the wording “superposition” instead. Without a means to separate then measure the individual molecules from an entangled state, the claim of entanglement is misleading. Therefore, I suggest distinguishing between the use of “entangled state”, which is ok in this case, and referring to the “entanglement” as a defined resource, which is not ok in this case. After these changes, I recommend the paper for publication in Nature Communications.

We have carefully implemented the changes requested by the Reviewer, by choosing the title “Tailoring the superradiant and subradiant nature of two coherently coupled quantum emitters” proposed by the Reviewer, by replacing “degree of entanglement” by “degree of superposition” throughout the manuscript, as suggested by the Reviewer, while using a few times “entangled state” as agreed by the Reviewer.

These changes requested by the Reviewer are highlighted in green in the revised version of the manuscript. We also have added the changes requested to comply with the editor policies and formatting requirements.